# When and How Does CLIP Enable Domain and Compositional Generalization?

Elias Kempf [* 1]  Simon Schrodi [* 1]  Max Argus [1]  Thomas Brox [1]

## Abstract

The remarkable generalization performance of contrastive vision-language models like CLIP is often attributed to the diversity of their training distributions. However, key questions remain unanswered: Can CLIP generalize to an entirely unseen domain when trained on a diverse mixture of domains (domain generalization)? Can it generalize to unseen classes within partially seen domains (compositional generalization)? What factors affect such generalization? To answer these questions, we trained CLIP models on systematically constructed training distributions with controlled domain diversity and object class exposure. Our experiments show that domain diversity is essential for both domain and compositional generalization, yet compositional generalization can be surprisingly weaker than domain generalization when the training distribution contains a suboptimal subset of the test domain. Through data-centric *and* mechanistic analyses, we find that successful generalization requires the learning of sufficiently shared representations in intermediate layers and circuits.

## 1. Introduction

Foundation models are considered a decisive step towards more generic AI models (Bommasani et al., 2021). For example, CLIP scaled the alignment of image-text pairs via a contrastive loss to millions of samples (Radford et al., 2021; Jia et al., 2021; Zhai et al., 2023). Unlike traditional classifiers from the ImageNet era, which often experience substantial performance drops under distribution shifts, CLIP demonstrates unprecedented generalization to "Out-of-Distribution (OOD)" data (Radford et al., 2021). However, what drives this improved OOD generalization?

Recent work has converged on the conclusion that CLIP's *diverse* training distribution is the primary factor driving its unprecedented generalization performance. For example, Fang et al. (2022) found that other factors such as language supervision, training data size, or the contrastive loss play only a minor role, while Nguyen et al. (2022) showed that data quality is more important than quantity. More recently, Mayilvahanan et al. (2025) demonstrated that CLIP's "generalization performance [...] drops to levels similar to what has been observed for ImageNet-trained models" (Mayilvahanan et al., 2025, p. 10) by limiting the diversity of (visual) domains[1] to a minimum, i.e., by removing all non-natural samples. This shows that the mixture of various (non-natural) domains plays an important role for CLIP's generalization, yet the underlying mechanisms remain unexplored. This brings us to our core research question:

*How does the mixture of diverse (visual) domains in the training data affect CLIP's generalization performance?*

In particular, we investigate under which circumstances CLIP can learn the object class invariances across the training domains with the aim to generalize to entirely unseen domains—a fundamental question about its *domain generalization* capability (Blanchard et al., 2011; Muandet et al., 2013; Gulrajani & Lopez-Paz, 2021). We also study questions about CLIP's *compositional generalization* (Hupkes et al., 2020; Wiedemer et al., 2023), which is believed to be an important factor of its generalization performance (Mayilvahanan et al., 2024; Udandarao et al., 2024) and a long-standing challenge of machine learning research. Adapting Szabó's (2012) classical example, we ask pictorially: Can CLIP, trained on natural images of cats and dogs along with sketches of cats, generalize to sketches of dogs?

To answer such questions, we construct fully controllable experimental conditions that allow precise and systematic manipulation of the domain mixtures and exposure to object classes in the training data (see Figure 1), while keeping all other variables, such as the CLIP model type, training process, and class distribution constant. Specifically, we augmented a base dataset consisting primarily of natural images, such as ImageNet-Captions (Fang et al., 2022), with non-natural samples from DomainNet (Peng et al., 2019),

---
[*]Equal contribution [1]University of Freiburg. Correspondence to: Elias Kempf <kempfe@cs.uni-freiburg.de>, Simon Schrodi <schrodi@cs.uni-freiburg.de>.

*Proceedings of the 42ⁿᵈ International Conference on Machine Learning*, Vancouver, Canada. PMLR 267, 2025. Copyright 2025 by the author(s).

---
[1]We refer to a domain as a group of images sharing a common style, such as natural images, sketches, paintings, etc.

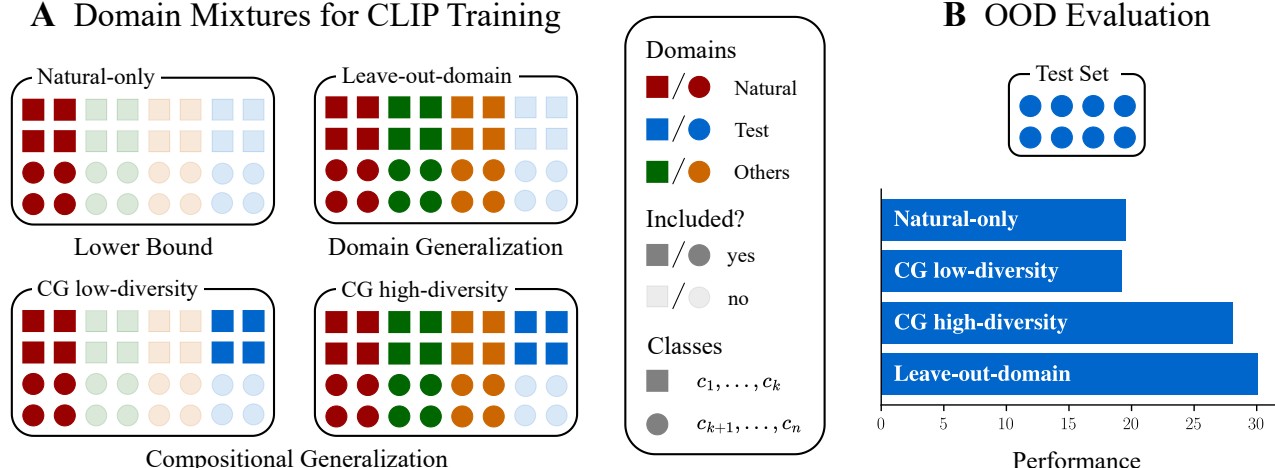

Figure 1: **Training data setups and CLIP's generalization performance under varying domain mixtures and class exposure. A**: We systematically varied the domain mixture and object class exposure in CLIP's training data across four controlled scenarios, while keeping other factors like dataset size and model choice constant. Specifically, we trained CLIP models on (1) *Natural-only*—mostly natural images (from classes ■, ●) to establish a lower bound on performance; (2) *Leave-out-domain*—a diverse set of domains (■, ●, ■, ●, ■, ●) excluding the test domain to assess CLIP's domain generalization; (3) *CG low-diversity*—natural images (■, ●) with some test domain classes (■) to evaluate its compositional generalization with limited domain diversity; and (4) *CG high-diversity*—a diverse mix of domains (■, ●, ■, ●, ■, ●) plus some test domain classes (■) to assess compositional generalization with higher domain diversity. **B**: CLIP models trained on diverse domains demonstrate stronger OOD generalization to held-out test domain classes (●), compared to models trained only on natural images or fewer domains. Notably, CLIP can perform as well or better even without exposure to some test domain classes during training.

including the domains Clipart, Infograph, Painting, Quickdraw, and Sketch. By systematically including subsets of these domains and their classes, we study questions about CLIP's domain generalization and compositional generalization capabilities. We complement these experiments with in-depth data-centric and mechanistic analyses to understand what changes in the CLIP model led to the improved generalization or failure thereof. Our experiments uncovered the following key findings:

- **Domain diversity improves generalization**: We reaffirm the intuition that diversity of domains in the training distribution is critical for both domain and compositional generalization. However, CLIP only *weakly* generalizes, as there remains a performance gap to a model that has seen similar samples during training.

- **Compositional generalization is challenging**: Surprisingly, including a domain in the training data does not always improve generalization to unseen classes within that same domain. However, ensuring sufficient class diversity within the test domain—ideally with no overlap with the queried classes in evaluation—along with high domain diversity, can significantly reduce the aforementioned performance gap.

- **Generalization requires sufficient feature and circuit sharing**: When CLIP generalizes well composi-

tionally, it shares more embedding features between different domains. However, CLIP sometimes fails to generalize to certain domains. We provide a twofold explanation: (1) the inputs from these domains lack shared input features and, as a result, (2) the model has limited shared intermediate features and circuits[2] between domains, constraining its ability to generalize effectively. We support this hypothesis through representational similarity analysis and introduce a related concept for circuits: *mechanistic similarity*, measuring similarity between circuits.

This work presents a systematic study of CLIP's domain and compositional generalization, unveiling both its capabilities and its limitations. For reproducibility, our code is available at https://github.com/lmb-freiburg/understanding-clip-ood.

## 2. Related Work

**OOD generalization of CLIP**   Unlike earlier models from the ImageNet era, CLIP models have been shown to exhibit remarkable OOD generalization (Radford et al., 2021; Geirhos et al., 2021). Subsequent work has further enhanced

---

[2]Interconnected internal model mechanisms/components for performing a specific computation or task (Olah et al., 2020).

this capability by introducing stylistic diversity or encouraging the learning of domain-invariant representation (Huang et al., 2023; Bose et al., 2024; Addepalli et al., 2024; Yu et al., 2024). These advances in OOD generalization have sparked research to uncover the underlying factors behind CLIP's strong OOD generalization.

Fang et al. (2022) showed that CLIP's *diverse* training distribution—rather than its dataset size, language supervision, or contrastive loss—is the primary factor of its OOD generalization. Similarly, Nguyen et al. (2022) found that dataset quality outweighs quantity. However, the precise characteristics of the training distribution contributing to CLIP's generlization performance remained unclear.

While high train-test similarity was initially believed to be a key factor, Mayilvahanan et al. (2024) found its impact to be smaller than expected. Instead, other factors, such as the class distribution, were shown to play a more important role (Wen et al., 2024). Moreover, caption richness has been found to enhance CLIP's robustness (Xue et al., 2024; Wen et al., 2024), and CLIP's loss fosters the learning of disentangled representations, facilitating the generalization to unseen attribute-object combinations (Abbasi et al., 2024). At the same time, other work revealed that CLIP exhibits behaviors resembling those of supervised classifiers. For example, CLIP fails to generalize when *all* non-natural images are removed from its training data (Mayilvahanan et al., 2025), and CLIP can be vulnerable to spurious correlations (Wang et al., 2024). While these works have significantly advanced our understanding of CLIP's OOD generalization, key questions remain. For example, "Can CLIP generalize to an entirely unseen domain?" (domain generalization), "Can CLIP generalize to unseen class-domain combinations?" (compositional generalization), and which factors contribute to such generalization?

**OOD generalization beyond CLIP**  The study of OOD generalization has been a focal point in the recent machine learning literature, covering various learning setups (refer to Table 2 of Gulrajani & Lopez-Paz (2021) for a comprehensive overview). In this work, we focus on two specific setups: domain generalization and compositional generalization. In domain generalization, models are trained on *multiple domains* and evaluated on an *entirely unseen domain* (Blanchard et al., 2011; Muandet et al., 2013; Gulrajani & Lopez-Paz, 2021). This is known as "learning from multiple environments" in the causality literature (Peters et al., 2016; Arjovsky et al., 2019; Arjovsky, 2019; Richens & Everitt, 2024). Compositional generalization, on the other hand, examines whether models generalize to unseen combinations of factors which were seen separately in training. Compositional generalization was recently studied for, e.g., (causal) generative models (Atzmon et al., 2020; Okawa et al., 2023; Wiedemer et al., 2023), object-centric models (Wiedemer et al., 2024), disentangled (Xu et al., 2022), and

(general) visual representation learning (Misra et al., 2017; Schott et al., 2022; Saranrittichai et al., 2022).

## 3. Problem Setup

Recent work highlighted that CLIP is trained on a substantial amount of non-natural images (Mayilvahanan et al., 2025, Table 2), but the role of these non-natural images in enabling CLIP's OOD generalization remains unclear. In this work, we aim to address this question by systematically analyzing the effect of different domain mixtures in CLIP's training data (see Figure 1), allowing us to study CLIP's ability to generalize to entirely unseen domains (*domain generalization*) and to novel combinations of known domains and classes (*compositional generalization*).

**Notations**  Let $D_0$ denote the base domain mostly consisting of natural images with image-text pairs $(I_i^0, T_i^0)$ (red in Figure 1). Further, we consider $m$ non-natural domains $D_r$ for $r \in \{1, \ldots, m\}$ with image-text pairs $(I_i^r, T_i^r)$ (blue, green, orange). Lastly, we consider the object classes $C = \{c_1, \ldots, c_n\}$ for the images $I$ which we divide into two disjoint subsets $C_1 = \{c_1, \ldots, c_k\}$ (squares ■) and $C_2 = \{c_{k+1}, \ldots, c_n\}$ (circles ●). We denote the subset of $D_r$ that contains only classes from $C'$ as $D_r^{C'}$.

**Training data setups**  Below, we specify the four training data setups; see Figure 1 for a visual overview.

- **Natural-only (lower bound)**: We train only on our base image domain $D_0$ (e.g., ImageNet-Captions (Fang et al., 2022), see Section 4 for further details), consisting (almost) exclusively of natural images of *all* classes $C$ (■, ●). This condition serves as our lower bound and mirrors the training distributions considered in Fang et al. (2022); Mayilvahanan et al. (2025).

- **Leave-out-domain (domain generalization)**: We train on a variety of domains but hold out the test domain $D_i$: $\bigcup_{j \neq i} D_j \cup D_0$ (■, ●, ■, ●, ■, ●); following the classical domain generalization learning setup (Blanchard et al., 2011; Muandet et al., 2013; Gulrajani & Lopez-Paz, 2021). Note we could also interpret this as *extrapolation* in certain cases, i.e., does CLIP generalize to data *outside* the domain coverage seen during training?

**Definition 3.1.** A model *compositionally generalizes* if it can accurately classify any new combinations of seen factors—here, classes and domains—not seen together during training.

Following this definition, we construct the training data setups for Compositional Generalization (CG) as follows:

- **CG low-diversity**: We train on the base domain $D_0$ and a subset of classes $C_1$ of the test domain $D_i$: $D_0 \cup D_i^{C_1}$ (■, ●, ■).

- **CG high-diversity**: We train on the base domain $D_0$, a diverse set of other domains $\mathbf{D}_{j \neq i} := \bigcup_{j \neq i} D_j$ containing *all* classes $C$, and a subset of classes $C_1$ of the test domain $D_i$: $D_0 \cup \mathbf{D}_{j \neq i} \cup D_i^{C_1}$ (■, ●, ■, ●, ■, ●, ■).

**Test data** Our test data for *all* training data setups consists of the *same novel* combinations of the classes $C_2$ of the test domain $D_i$: $D_i^{C_2}$ (●). By keeping the test set fixed throughout all conditions, we ensure comparability across these setups.

## 4. Experimental Setup

**Datasets** We used either ImageNet-Captions (Fang et al., 2022), CC3M (Sharma et al., 2018), or CC12M (Changpinyo et al., 2021) as our base image-text datasets $D_0$ (red in Figure 1). Captions in ImageNet-Captions are constructed using the title, tag, and description (if provided) to yield maximal descriptiveness. To mitigate the influence of class distribution shift, we augmented the base datasets with natural samples from DomainNet-Real (Peng et al., 2019).

For the domain-specific image-text pairs $D_r$ (blue, green, orange in Figure 1), we used DomainNet's non-natural domains: Clipart, Infograph, Painting, Quickdraw, and Sketch (Peng et al., 2019). Since DomainNet provides no captions, we created captions by using *domain-invariant* templates (e.g., `an image of a {class}`) or *domain-specific* templates (e.g., `a {domain} of a {class}`); see Appendix A.2 for further details. We used comparable final training dataset sizes across our different training conditions; refer to Appendix A.3 for details. Finally, the class choices for $C_1$ and $C_2$ are provided in Appendix A.4.

**Evaluation of CLIP models** We evaluated CLIP models in the classical zero-shot classification setting across all DomainNet classes $C$ using the standard OpenAI templates (Radford et al., 2021), extended with templates for the missing domains of DomainNet; see Appendix A.5 for further details. To mitigate the effect of class imbalance, we calculated the *balanced* top-1 accuracy, which we will hereon refer to as top-1 accuracy for brevity.

## 5. When Does CLIP Exhibit Domain and Compositional Generalization?

In this section, we trained CLIP models on our systematically constructed training data setups, as described in the previous sections and illustrated in Figure 1 (refer to Appendix A.6 for training details), to investigate *when* CLIP achieves domain generalization and compositional generalization. Figure 2 summarizes the results for CLIP models with ResNet-50 vision encoder and trained on ImageNet-

Captions as base dataset. We discuss the results and key findings below.

To ensure the validity of our results and findings, we validated them across several alternative choices: (1) base datasets (CC3M, CC12M), (2) vision encoders (ViT-S-32, Swin-T), (3) contrastive loss choices (SigLIP; Zhai et al., 2023), and (4) vision-language pre-training methods (BLIP; Li et al., 2022). These additional results, which are consistent with those in Figure 2, are provided in Appendix B.1.

**The role of domain diversity** By constructing domain mixtures and controlling for all other factors, such as dataset size or model choice, we are able to isolate the impact of domain diversity in Figure 2. In particular, Figure 2 reaffirms the hypothesis that domain diversity is a key factor in enhancing CLIP's generalization: CLIP achieves both significantly better domain and compositional generalization in settings with high domain diversity (Leave-out-domain, CG high-diversity) compared to settings with low domain diversity (Natural-only, CG low-diversity).

**Does each domain contribute equally?** While these results demonstrate that domain diversity is critical for the (compositional) generalization of CLIP, they do not provide insights into the importance of each domain individually, since all domains are added at once. Thus, we successively added domains to CG low-diversity until arriving at CG high-diversity to assess their importance.

Figure 3 shows that some domains contribute more strongly to generalization, while other domains can even slightly reduce generalization performance on the test domain. Intriguingly, the impact of a domain can change, depending on which other domains are already included. For example, in Figure 3a, while Sketch and Quickdraw are the most beneficial domains to include initially for improving generalization performance on Clipart, once Sketch is added, it is more effective to first include the other domains (Painting, Infograph) before adding Quickdraw. We hypothesize that this is due to the visual similarity between Sketch and Quickdraw—both consist of black-and-white/gray images with object contours—which may lead the model to learn similar features. Once such features are learned from one domain, the benefit of adding the other diminishes. Further investigation of this observation is left for future work.

Overall, these observations suggest that while domain diversity is generally beneficial for generalization, the specific relationship between the added domain(s) and the test domain plays a critical role.

> **Finding 1:** Domain diversity enhances domain generalization and compositional generalization. However, the relationship between domains also matters.

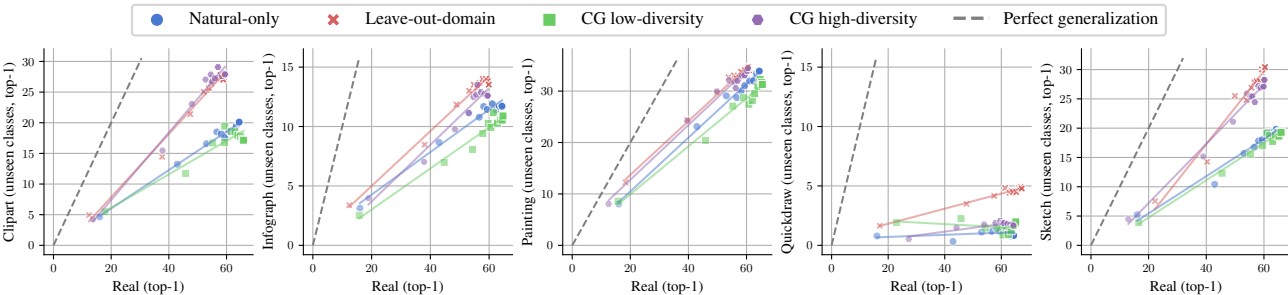

Figure 2: **High diversity domain mixtures exhibit improved effective robustness.** Each point represents the average balanced top-1 accuracy over three consecutive training epochs and three seeds, with higher opacity indicating later training epochs. High diversity domain mixtures (Leave-out-domain (red), CG high-diversity (purple)) have consistently higher generalization performance than their low diversity counterparts (Natural-only (blue), CG low-diversity (green)). These gains are especially pronounced in the Clipart and Sketch domains (leftmost and rightmost). However, for the Quickdraw domain (4th from left), generalization fails even in the high diversity settings—a limitation we will further investigate in Section 6.2. Other evaluation metrics (balanced top-5 accuracy and $F_1$ score) yield consistent results (see Figure 8 in Appendix B).

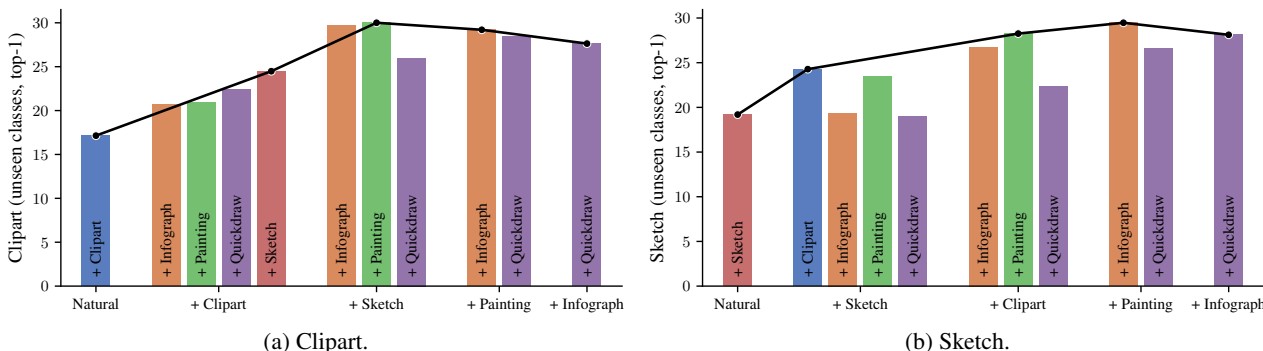

Figure 3: **Certain domains contribute more strongly to generalization performance than others.** When domains are added successively from CG low-diversity to CG high-diversity (from left to right), some domains significantly improve generalization performance (balanced top-1 accuracy) on the unseen classes of the test domain, while others provide only small gains or even slightly degrade it.

**How close is CLIP to the maximally achievable performance?** In our previous experiments, we observed that CLIP generalizes well given sufficient domain diversity. However, how close is CLIP to the maximally achievable performance if it had been trained on class-specific samples from the test domain?

Table 1 shows that, even in high diversity settings with better generalization, a gap to this upper bound performance remains. We analyzed the number of test class samples required to close this gap in Appendix B.3. We found that the number of required samples tends to scale linearly.

**Achieving good compositional generalization is challenging** While high diversity improves generalization, one may intuitively expect CLIP to generalize better to unseen classes within a test domain if it has seen some other classes of that domain during training. However, Figure 2 surpris-

ingly reveals the opposite: CLIP models trained on a subset of classes from the test domain (CG low-/high-diversity) are often slightly outperformed by models that have not seen the test domain at all (Natural-only, Leave-out-domain).

> **Finding 2:** Compositional generalization can be weaker than domain generalization.

This finding is surprising, since the test domain is entirely unseen and domain generalization can require extrapolation. In contrast, compositional generalization is expected to perform better as it has partial exposure to that domain. However, our results suggest that this intuition may not always hold.

To better understand this, we investigated the role of the test domain's chosen classes for training and the ones that are queried during evaluation. For example, CLIP may learn the shortcut that all sketches belong to the subset of the

Table 1: **There is a performance gap when CLIP has seen class-specific samples of the test domain.** With sufficient domain diversity, CLIP generalizes well to unseen the classes $C_2$ of the Clipart and Sketch test domains $D_i$. However, even in these cases, there remains a gap in balanced top-1 accuracy compared to models also trained on domain-specific samples of the classes, i.e., $D_i^{C_2}$. Appendix B.3 provides the results of the other domains.

| Training data setup (Figure 1) | Clipart | Sketch |
|---|---|---|
| Leave-out-domain | 27.4 | 30.1 |
| CG high-diversity | 27.6 | 28.1 |
| w/ classes $C_2$ (upper bound) | 36.6 (+9.0) | 44.3 (+16.2) |

Table 2: **Seeing a subset of classes of the test domain can worsen compositional generalization.** Including a subset of sketches from DomainNet (CG low-/high-diversity) slightly decreases balanced top-1 accuracy on the unseen sketch classes $C_2$ compared to not seeing that domain at all (Leave-out-domain). However, adding sketches of classes that do not overlap with DomainNet's classes, instead improves compositional generalization performance, suggesting that compositional generalization can be limited by (only) partial, suboptimal inclusion of the test domain.

| Training data setup (Figure 1) | Sketch |
|---|---|
| Natural-only | 19.5 |
| Leave-out-domain | 30.1 |
| CG low-diversity | 19.2 |
| w/ sketches of non-queried classes only | 27.1 (+7.9) |
| CG high-diversity | 28.1 |
| w/ sketches of non-queried classes only | 36.9 (+8.8) |

seen classes, which becomes wrong for the unseen classes queried in evaluation. To test this hypothesis, we replaced DomainNet's sketches from the classes $C_1$ in our previous CG settings with sketches from a set of classes $C_3$ which is disjoint from $C_1 \cup C_2$ and *not* queried in evaluation. For this, we used ImageNet-Sketch (Wang et al., 2019) and excluded all ImageNet classes that overlap with the classes from DomainNet; refer to Appendix A.3 for details on the class overlap.

Table 2 confirms that including sketches of non-queried classes improves compositional generalization. However, it also highlights a failure mode: while compositional generalization can work for CLIP (to some extent, c.f., Table 1), partial exposure to classes of the test domain that overlap with the classes queried in evaluation significantly worsens compositional generalization. We further analyzed the severity of this overlap on compositional generalization performance in Appendix B.2.

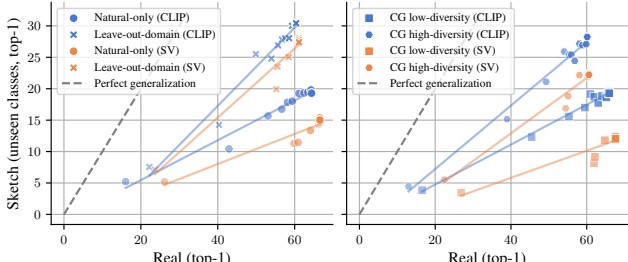

Figure 4: **Effective robustness of CLIP vs. supervised classifiers.** Similar to CLIP, the robustness of supervised classifiers also increases with the domain diversity of the training data. However, CLIP consistently shows superior balanced top-1 accuracy. Refer to Appendix B.4 for the results on the remaining domains.

Table 3: **Domain diversity increases feature sharing in the embeddings.** We report the percentage point increase in top-k shared SAE features, averaged over classes $c \in C_2$ and $k \in \{5, 10, 15, 20\}$ (see Appendix C.1 for details on the computation of shared features), when comparing low diversity (Natural-only) to high diversity (Leave-out-domain, CG high-diversity). Note that we used the CLIP models using CC12M as base dataset, since we found that SAEs extracted poor features for the models trained on ImageNet-Captions.

| | Clipart | Sketch |
|---|---|---|
| Natural-only $\rightarrow$ Leave-out-domain | +7.1 | +4.1 |
| Natural-only $\rightarrow$ CG high-diversity | +6.9 | +3.4 |

**The role of language supervision** To investigate the influence of language supervision, we replicated our previous experiments with a supervised ResNet50 classifier. Specifically, we trained the classifier on the combined class distribution of ImageNet and DomainNet. Refer to Appendix A.7 for further training details.

Figure 4 shows that generalization performance of supervised classifiers increases with domain diversity, similar to CLIP. However, CLIP consistently exhibits higher performance compared to the classifiers, which can be attributable to caption richness (Xue et al., 2024; Wen et al., 2024).

## 6. Why Does CLIP (Not) Generalize?

In this section, we investigate which changes in the CLIP model led to domain and compositional generalization, or lack thereof, observed in the previous section.

### 6.1. The Role of Visual Embeddings

Intuitively, we would expect CLIP to share more features in its visual embeddings across domains as generalization im-

Table 4: **Domain-specific captions do not explain CLIP's poor generalization performance to unseen Quickdraw classes in the CG high-diversity setting.** When all domain-specific captions are replaced (second row), we observe increased alignment in the visual embeddings (Figure 5b); yet, this does not translate into improved generalization performance, as measured by balanced top-1 accuracy.

| Captions | | Classes | |
|---|---|---|---|
| domain-specific | domain-invariant | seen | unseen |
| 50% | 50% | 50.7 | 1.7 |
| 0% | 100% | 46.2 | 0.7 |

proves. To test this hypothesis, we applied an unsupervised dictionary learning technique, i.e., Sparse Autoencoders (SAEs) (Bricken et al., 2023; Huben et al., 2024), to extract interpretable features from CLIP's visual embeddings $\mathbf{a} \in \mathbb{R}^p$:

$$\text{SAE}(\mathbf{a}) := (g \circ \phi \circ f)(\mathbf{a}), \qquad (1)$$

where $\phi$ is a ReLU non-linearity, and $f$ and $g$ are the linear encoder with weights $\mathbf{W}_f \in \mathbb{R}^{p \times h}$ and decoder with weights $\mathbf{W}_g \in \mathbb{R}^{h \times p}$, respectively. We trained the SAE with an $L_2$ reconstruction loss and $L_1$ sparsity regularization. Refer to Appendix C.1 for further technical details.

After extracting interpretable SAE features, we measured the percentage overlap of the top-$k$ most important SAE features between domain pairs for each class $c \in C_2$. Specifically, to identify important SAE features, we counted how often each SAE feature appeared among the top-20 highest-activating SAE features across domain-class combinations. For each class, we then calculated the percentage overlap of the top-$k$ (with $k \in \{5, 10, 15, 20\}$) of these top-20 most frequent SAE features between the test domain and each of the other domains. These percentage overlaps were finally averaged over all values of $k$, all classes in $C_2$, and all pairs that include the test domain. Further details on this computation and pseudocode are provided in Appendix C.1.

As shown in Table 3, CLIP models trained on more diverse data—which generalize better—exhibit more cross-domain feature sharing in their visual embeddings.

> **Finding 3:** CLIP shares more features across domains in its embeddings as generalization improves.

### 6.2. Why Does CLIP Sometimes Fail To Generalize?

Figure 2 shows that CLIP typically generalizes well compositionally with high domain diversity, except for Quickdraw. It is tempting to attribute this solely to the unique characteristics of Quickdraw images (Appendix A.1). However, the first row of Table 4 highlights that CLIP can correctly clas-

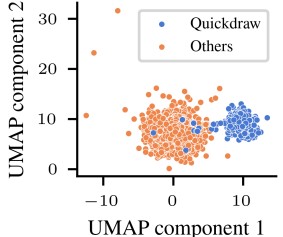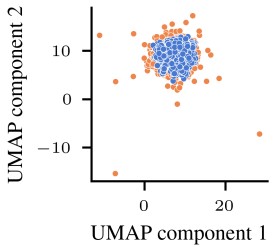

(a) Domain-invariant and specific captions.

(b) Only domain-invariant captions.

Figure 5: **The separation of the visual embeddings between Quickdraw and the images of other domains is due to domain information in captions.** However, the better alignment of the visual embeddings does not improve compositional generalization (Table 4).

sify the Quickdraw images of seen classes, indicating that is has learned something useful for the Quickdraw domain, yet CLIP performs poorly on images of unseen classes.

**Are domain-specific captions the cause?** We initially hypothesized that using domain-specific captions—our training included both domain-invariant and domain-specific captions (Section 4)—might have inadvertently led CLIP to prioritize uniformity over alignment in its loss function. We suspect this occurs due to the stark visual differences between Quickdraw images and those from other domains, making alignment challenging. In order to minimize the total loss in spite of this, CLIP may adopt a short-cut: prioritizing uniformity, aligning Quickdraw image embeddings with the text embeddings of the domain-specific captions, while sacrificing alignment with domain-invariant captions. Consequently, we would expect a clear separation between Quickdraw image embeddings and those from other domains—and indeed, we observe this (Figure 5a). However, this separation may come at a cost, limiting CLIP's ability to generalize across the Quickdraw domain, particularly to unseen classes.

As a remedy, we trained CLIP using only domain-invariant captions. Although the visual embeddings are now better aligned (Figure 5b), generalization performance remains poor (second row of Table 4). Thus, the domain invariance or specificity of captions cannot solely explain CLIP's poor generalization performance on unseen classes from Quickdraw.

**The role of shared intermediate features and circuits** While visual embeddings appear aligned in Figure 5b, it does not necessarily imply that the computations leading up to these embeddings are also aligned. Thus, we hypothesize that *insufficient sharing of intermediate representations and circuits* (Hohman et al., 2019; Olah et al., 2020) within

CLIP's vision encoder may explain its poor generalization to the unseen classes. For example, if CLIP learns a separate circuit for Quickdraw images instead of sharing sufficient functionality across the domains, such a circuit may work for seen classes but will fail on unseen classes, since the unseen classes of the test domain can only be inferred with the help of the other domains.

To test our hypothesis, we used tools from representational similarity analysis to evaluate intermediate representations. Additionally, we introduce a novel, related concept for circuits: *mechanistic similarity*, which measures the similarity between circuits.

**Representational similarity analysis**  To assess whether CLIP learns similar intermediate representations for the same class across domains, we measured representational similarity using Center Kernel Alignment (CKA) (Kornblith et al., 2019) with the unbiased Hilbert-Schmidt Independence Criterion (HSIC) estimator (Song et al., 2012); refer to Appendix C.2 for technical details.

**Mechanistic similarity analysis**  While representational similarity captures alignment at the representational level, it does not reveal whether similar computations are implemented internally. To address this, we introduce a measure of *mechanistic similarity*, which quantifies the similarity of the underlying circuits, i.e., the sets of model components and their interactions responsible for a specific prediction. Our analysis proceeds in two steps:

1. **Circuit discovery**: For each class-domain pair, we extracted a circuit represented as a directed acyclic graph. We first identified the top-$k$ model components—specifically, axis-aligned neurons—based on their indirect effect[3] (Pearl, 2001) on the model's predictions, following prior work (Vig et al., 2020; Schrodi et al., 2022; Meng et al., 2022; Marks et al., 2025). These components serve as the nodes of the circuit. Next, for each component, we identified its $k'$ most influential predecessors (also based on indirect effect), which define the edges of the circuit. Refer to Appendix C.3 for further details on circuit discovery.

2. **Circuit similarity**: We then compared circuits across domains for each class—for example, comparing the circuit for "dog" of Quickdraw images vs. those of Clipart, Infograph, Painting, Real (natural images), and Sketch images. To quantify similarity, we computed the layer-wise Jaccard index of the top-$k$ components, capturing node overlap. To assess deeper structural similarity, we applied the normalized Weisfeiler-Lehman subtree graph kernel (Shervashidze et al., 2011), which

---

[3]Intuitively, the indirect effect quantifies how much each component contributes to the model's prediction.

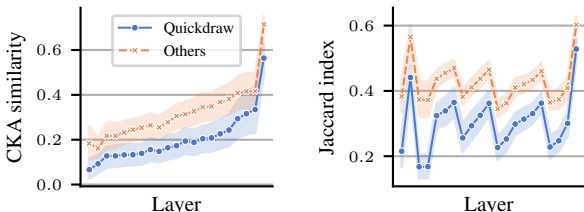

(a) Representational similarity.  (b) Amount of shared neurons.

|  | Quickdraw | Others |
|---|---|---|
| Weisfeiler-Lehman similarity (↑) | 0.218 | 0.28 |

(c) Circuit similarity.

Figure 6: **CLIP's intermediate representations and circuits show a significant difference between Quickdraw and the other domains, even though final visual embeddings can be aligned.** We used the CLIP model trained on only domain-invariant captions from Figure 5b and Table 4 (second row). Scores are averaged over classes and higher scores mean higher representational similarity (a), more shared neurons (b), or more similar circuits (c).

captures more complex structural and hierarchical similarities. More details are provided in Appendix C.3.

**Result**  Figure 6 confirms that the Quickdraw domain noticeably differs from the other domains, both in terms of representational and mechanistic similarity, supporting our hypothesis that (compositional) generalization requires sufficient sharing of intermediate features and circuits.

> **Finding 4:**  Sufficient sharing of intermediate features and circuits is crucial for generalization to succeed.

## 7. Discussion

**Compositional generalization**  Our results indicate that CLIP has a (weak) ability for compositional generalization, which is influenced by the composition of its training data. Specifically, CLIP's compositional generalization performance worsens when a subset of classes of the test domain, that is queried during evaluation, is seen during training (Table 2). However, by including only classes that do not overlap with the queried classes in the training data, CLIP's compositional generalization significantly narrows the gap to the maximally achievable performance. In particular, performance on unseen sketch classes significantly improved from 19.5% (Natural-only) to 36.9% (CG high-diversity w/ non-queried classes only, Table 2), approaching the maximally achievable performance of 44.3% (Table 1). This result supports the hypothesis that compositional generalization could be indeed a driver behind CLIP's generalization.

Our results also allow us to reinterpret the main experiment from Mayilvahanan et al. (2024, Table 1). In their experiment, the maximally achievable performance corresponds to the unaltered LAION-200M dataset, while the pruned versions of LAION-200M—where, e.g., (near) duplicates of all ImageNet-Sketch classes were removed—can be interpreted analogous to the curated compositional generalization setting with high diversity from above. They observed only a slight deterioration in performance despite these exclusions. Our result suggests that compositional generalization could be the key contributor of the sustained performance observed here, though further analysis is required to exclude other (unknown) factors.

**Rethinking CLIP's OOD generalization**   Recent studies have concluded that CLIP's OOD generalization is largely driven by its vast and diverse training distribution (Fang et al., 2022; Mayilvahanan et al., 2025). Our experiments reaffirm this: While CLIP can *weakly* generalize (Figure 2), a generalization gap persists when compared to models that have seen class samples of the test domain(s) during training (Table 1). We believe it is worth studying how narrow this generalization gap can become and if it can ever be closed without access to (near) duplicate samples during training—both from empirical and theoretical perspectives.

**Mechanistic similarity analysis**   Mechanistic similarity analysis (Section 6.2) is a general framework that first discovers circuits and then measures their similarity. In future work, we plan to apply this framework to other problems, e.g., to investigate whether multi-lingual LLMs process different languages in similar or very different ways.

**Larger dataset sizes**   In this work, we disentangled the effects of dataset size and domain diversity—two factors that are typically confounded—and showed that domain diversity is a key driver of CLIP's generalization. However, dataset size alone may also impact CLIP's generalization performance.

At the time of writing, to the best of our knowledge, no large-scale domain datasets comparable to the size and diversity of DomainNet exists. To address this, future work could filter a large, diverse dataset, such as LAION (Schuhmann et al., 2021), by domains. For example, the data filtering approach proposed by Mayilvahanan et al. (2025), which classifies images into natural and non-natural, could be extended to classify various visual domains. Note that the existing filtered natural-only subset of LAION-200M curated by Mayilvahanan et al. could be directly adopted as the base dataset $D_0$.

Although, this would help overcome the scarcity of domain-specific data, we lack the computational resources to conduct such large-scale experiments. However, we are confi-

dent that our findings remain robust across dataset sizes, as demonstrated by consistent results in all three base datasets, with sizes ranging from approximately 0.5 M to 10 M samples, in Appendix B.1.

**Dataset quality**   Besides diversity, the quality of the datasets used for training also likely plays a crucial role in achieving robust generalization. For example, Nguyen et al. (2022) showed that combining multiple datasets does not necessarily result in improved robustness; in fact, the robustness of the best-performing dataset can even be diluted by lower-quality datasets. To date, dataset quality remains only loosely addressed in much of the current literature, largely due to the difficulty to quantify it. Thus, we leave further investigation of this aspect for future work.

**Carbon emission estimate**   We conducted our experiments mainly on NVIDIA RTX 2080 GPUs and estimated the total GPU hours to be approximately 25 000. With a carbon efficiency of 0.321 kgCO$_2$eq/kWh[4], total emissions are estimated to be about 1 725 kgCO$_2$eq, using the Machine Learning Impact calculator (Lacoste et al., 2019).

## 8. Conclusion

In this work, we analyzed when CLIP generalizes to unseen domains and subsets thereof using systematically created training data setups (see Figure 1), identifying domain diversity as a prerequisite for both domain and compositional generalization. We showed that compositional generalization can fail in certain scenarios, i.e., even perform worse than domain generalization. We supported these findings with in-depth data-centric experiments and mechanistic analyses, offering insights into the internal workings that make generalization succeed or fail.

## Impact Statement

This paper presents work whose goal is to advance the field of Machine Learning. There are many potential societal consequences of our work, none which we feel must be specifically highlighted here.

## Acknowledgements

This research was funded by the the German Research Foundation (DFG) under grant numbers 417962828, 539134284, and 499552394 (SFB 1597).

We would like to thank Evgenia Rusak, Thaddäus Wiedemer, and Prasanna Mayilvahanan for an insightful discussion.

---

[4]Estimated carbon efficiency of the German power grid in 2024 according to https://www.nowtricity.com/country/germany/.

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

# A. Further Details for Section 4

## A.1. DomainNet Examples

Figure 7 visualizes four examples from all six image domains (Clipart, Infograph, Painting, Quickdraw, Real, Sketch) from DomainNet (Peng et al., 2019). Some domains are more visually similar than others. For example, sketches and Quickdraw images are typically gray, while paintings often contain a similar level of image detail similar to that of Real (natural) images.

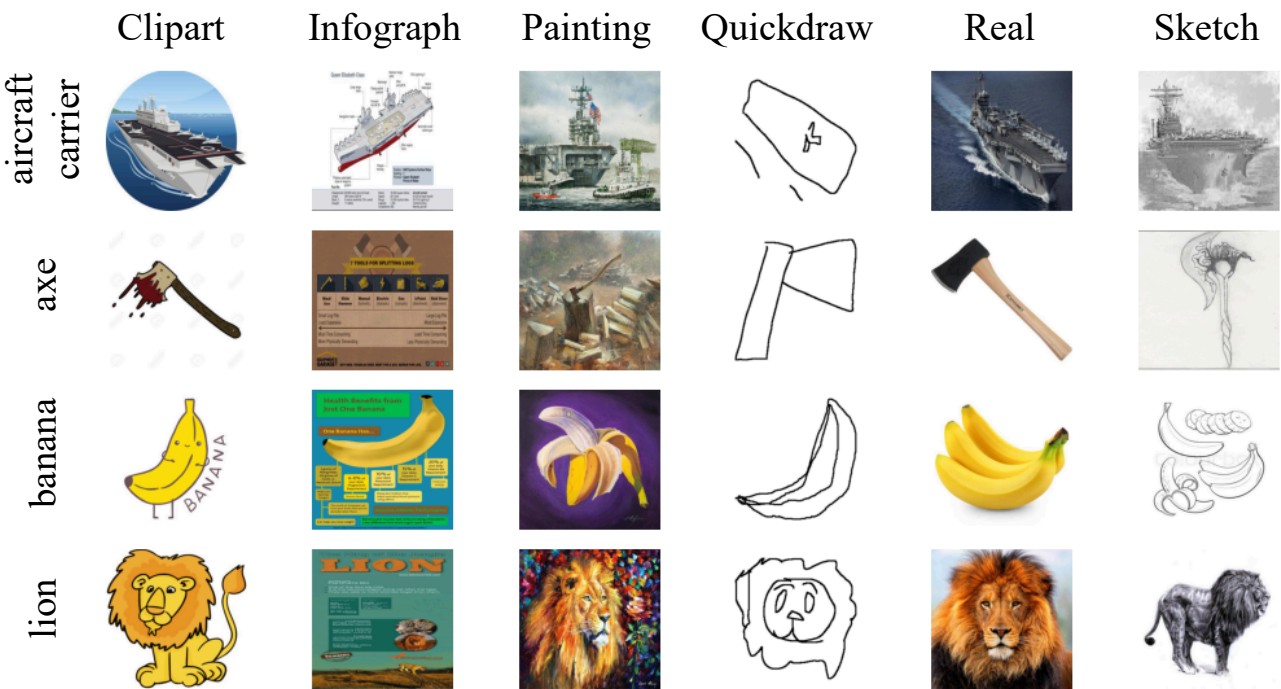

Figure 7: **Random examples across the six domains of DomainNet.**

## A.2. DomainNet Training Captions

The DomainNet dataset (Peng et al., 2019) does not include captions. Therefore, we created captions using the class names to train CLIP models on DomainNet data. For this, we used prompt templates similar to Radford et al. (2021). Specifically, we used the following templates:

- "a {domain} of a {class}.",

- "a {class} {domain}.",

- "a {domain} depicting a {class}.",

- "a {class} depicted in a {domain}.",

- "a {domain} showing a {class}.",

- "a {class} is visible in a {domain}.".

For each DomainNet image sample, we randomly sampled one of the templates and inserted the corresponding class name for the placeholder {class}. For the {domain} placeholder, we randomly chose with equal probability a generic, domain-invariant term, such as *image* or *picture*, or a domain-specific term, such as *clipart* or *painting*. The terms are provided in Table 5.

Table 5: **Generic, domain-invariant and domain-specific terms.** Generic, domain-invariant terms are shared across domains, while domain-specific terms can be either its domain name or a synonym.

| Domain | Terms |
|---|---|
| Generic | image, picture |
| Clipart | clipart, illustration |
| Infograph | infograph, informational chart |
| Painting | painting, art |
| Quickdraw | quickdraw, doodle |
| Real | photo, snapshot |
| Sketch | sketch, drawing |

### A.3. Further Details on the Training Data Construction

**Dataset Construction**   We created our training datasets based on a base dataset $D_0$ (i.e., ImageNet-Captions, CC3M, or CC12M) which provides a large collection of (mostly) natural images along with corresponding language descriptions. To mitigate effects stemming from to class shifts, we augmented the base dataset with samples from DomainNet-Real, which consists of natural images. Finally, we created different domain mixtures (see Figure 1) by incorporating subsets of samples from various non-natural domains of DomainNet $D_r$ with $r \in \{$Clipart, Infograph, Painting, Quickdraw, Sketch$\}$ into the training data, as outlined in Section 3.

**Subsampling**   To ensure fair performance comparisons between the different training data setups (see Figure 1), we applied subsampling to maintain comparable final dataset sizes when adding additional domains to the training data. We designed our subsampling method to preserve the original data distribution as much as possible. Specifically, if domain $D_i$ contained twice as many samples as domain $D_j$ before subsampling, this ratio remained approximately constant afterward. Likewise, the class distribution within each domain was preserved as much as possible.

Note that the non-natural domains in DomainNet vary significantly in size, e.g., Quickdraw has more than three times as many samples as Clipart. Thus, we chose to only keep dataset sizes fixed within the same test domain. That is, for a given test domain, CG low-diversity, CG high-diversity and Leave-out-domain datasets have the same size. However, dataset sizes may differ across test domains (i.e., CG low-diversity settings for different test domains are not necessarily of equal size). Since the Natural-only lower bound is independent of the choice of the test domain, it contains slightly fewer samples than the other mixtures.

**Joining the Class Distributions of ImageNet and DomainNet**   Both ImageNet-Captions and DomainNet provide class labels for training supervised classifiers. However, their class distributions differ significantly in diversity and granularity. ImageNet has nearly three times as many classes as DomainNet and is more fine-grained. For example, ImageNet distinguishes over 100 different dog breeds, whereas DomainNet has only a single dog class.

To address this, we created a mapping from ImageNet classes to DomainNet classes. Each ImageNet class was either mapped to a single DomainNet class or left unmatched, and multiple ImageNet classes could be mapped to the same DomainNet class. We constructed this mapping manually based on class names, the WordNet hierarchy (Miller, 1995), and the NAVIGU image explorer (https://navigu.net/#imagenet, Barthel et al. (2023)), ensuring that only semantically valid mappings were retained.

Our final mapping assigned 450 ImageNet classes to DomainNet, including 147 one-to-one mappings. Using this mapping, we merged the class distributions of ImageNet and DomainNet. Specifically, we relabeled the 450 mapped ImageNet classes with their corresponding DomainNet labels.

### A.4. Choice of Classes $C_2$

We carefully chose the subset $C_2$ in a way that the classes are diverse and not biased towards any spurious features (e.g., color). We considered the 147 classes, with a one-to-one match in ImageNet (see Appendix A.3), as the possible candidates for $C_2$. We selected about 10% of these candidates, i.e., 15 classes. For the selection process, we randomly sampled from

the set of candidates. To ensure that we adequately covered the different super-categories of DomainNet (e.g., furniture, mammal, tool), we kept only the first random sample for each category, rejecting further samples from the same category. We also manually rejected some samples if we considered them to be too similar to our existing selection. Our final selection of classes is

$$C_2 = \{\text{aircraft carrier, axe, banana, barn, bed, candle, lion, mountain,}$$
$$\text{necklace, penguin, pizza, saxophone, television, tractor, traffic light}\}. \tag{2}$$

and $C_1$ are the remaining 330 classes of DomainNet.

### A.5. DomainNet Evaluation Prompts

We evaluated the zero-shot performance of our CLIP models using the OpenAI templates from Radford et al. (2021). Since Painting and Sketch templates are already contained, we added the templates for the missing domain names:

- "a clipart of the {class}.",
- "a clipart of a {class}.",
- "an infograph of the {class}.",
- "an infograph of a {class}.",
- "a quickdraw of the {class}.",
- "a quickdraw of a {class}.".

Following Radford et al. (2021), we created zero-shot weights for all 345 DomainNet Classes from these templates by taking the class-wise average over the text embeddings of all templates (marginalization to obtain the "true" object embedding) and normalizing afterwards. Formally, let $c \in \{c_1, \ldots, c_n\}$ be a class, $T$ be the set of templates, $t_c$ a template with the name of class $c$ inserted, and $g$ be the text encoder of our CLIP model. Assuming that the text encoder $g$ produces $L_2$-normalized embeddings, we compute the zero-shot weights of class $c$ as:

$$\mathbf{w}_c = \frac{\frac{1}{|T|} \sum_{t \in T} g(t_c)}{\left\| \frac{1}{|T|} \sum_{t \in T} g(t_c) \right\|_2}. \tag{3}$$

### A.6. CLIP Training Details

Following Fang et al. (2022), we trained CLIP models with an embedding size of 1024 with ResNet-50 (He et al., 2016) and a transformer text encoder (Vaswani et al., 2017) (12 layers with a width of 512, 8 attention heads, and context length of 77). We trained the models for 32 epochs with a batch size of 1024 with AdamW (learning rate of 0.001, $\beta_1 = 0.9$, $\beta_2 = 0.999$, $\epsilon = $ 1e-8, weight decay of 0.2) and cosine annealing learning rate scheduling with 500 warmup steps. We used the code from OpenCLIP (Cherti et al., 2023) (https://github.com/mlfoundations/open_clip, License: custom) for our training implementation, including OpenCLIP's default augmentations.

### A.7. Supervised Classifier Details

Similar to Fang et al. (2022), we trained the supervised ResNet-50 classifiers (He et al., 2016) for 90 epochs with a batch size of 256 using SGD with Nesterov momentum, weight decay of 1e-4, momentum of 0.9, initial learning rate of 0.01 (they used 0.1) with step-wise decay by 0.1 at epochs 30, 50, and 70. We used the same image augmentations as for our CLIP models. We trained our supervised classifiers on the 895 classes—550 unmatched ImageNet classes combined with the 345 DomainNet classes—see Appendix A.3 for details

## B. Additional Results For Section 5

In Figure 2, we compared the effective robustness trends of our different training setup across different domains. Table 6 shows the final balanced top-1 test accuracy (averaged over three runs) of our CLIP models with ImageNet-Captions as the base dataset and ResNet-50 as the vision encoder.

Table 6: **CLIP robustness results for the four training data setups (see Figure 1).** We repeated CLIP trainings three times and evaluated the model's balanced top-1 accuracy on the unseen classes of the test domain. CLIP generalizes better with higher domain diversity. Intriguingly, CLIP achieves superior performance when not seeing the domain at all vs. seeing a subset of it (c.f., Table 2). However, while diversity substantially improves CLIP's generalization performance, there remains a performance gap to a model that has seen similar samples to the classes $C_2$ (c.f., Table 1). Figure 2 shows the respective effective robustness plots.

| Data composition (Figure 1) | Clipart | Infograph | Painting | Quickdraw | Sketch |
|---|---|---|---|---|---|
| Natural-only | $20.3 \pm 0.7$ | $11.8 \pm 0.2$ | $34.1 \pm 1.4$ | $0.8 \pm 0.1$ | $19.5 \pm 0.7$ |
| Leave-out-domain | $27.4 \pm 1.2$ | $13.6 \pm 0.6$ | $33.8 \pm 0.5$ | $4.8 \pm 1.1$ | $30.1 \pm 1.4$ |
| CG low-diversity | $17.1 \pm 1.2$ | $10.8 \pm 1.0$ | $31.4 \pm 0.1$ | $2.0 \pm 0.9$ | $19.2 \pm 2.0$ |
| w/ classes $C_2$ (upper bound) | $37.2 \pm 0.8$ | $21.5 \pm 1.8$ | $45.5 \pm 0.8$ | $56.0 \pm 0.6$ | $50.3 \pm 1.2$ |
| CG high-diversity | $27.6 \pm 1.5$ | $12.7 \pm 2.0$ | $34.6 \pm 1.2$ | $1.7 \pm 0.7$ | $28.1 \pm 0.8$ |
| w/ classes $C_2$ (upper bound) | $36.6 \pm 1.0$ | $18.8 \pm 0.3$ | $41.8 \pm 1.4$ | $51.5 \pm 1.9$ | $44.3 \pm 0.8$ |

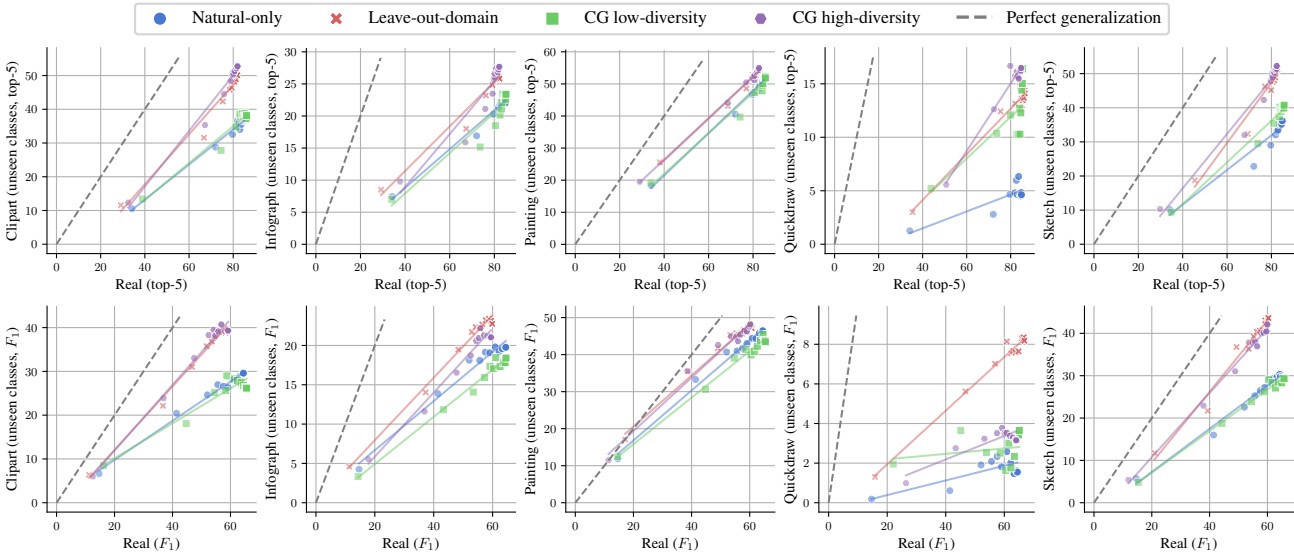

Figure 8: **Effective robustness for additional evaluation metrics.** We re-visualized the results from Figure 2 using balanced top-5 accuracy (first row) and macro $F_1$ score (second row). Our findings remain consistent across both metrics.

**Other evaluation metrics**    We also evaluated our CLIP models using balanced top-5 accuracy and macro $F_1$ score to ensure our findings remain consistent across different evaluation metrics. Figure 8 shows the corresponding effective robustness plots.

### B.1. Validity of the Results across Architecture, Dataset, and Loss Choices

We conducted the experiments from the main text (Figure 2 and Table 6) using ImageNet-Captions as the base dataset and a ResNet-50 vision encoder. To ensure the consistency of our findings across different base datasets, vision encoder architectures, and contrastive loss functions, we performed additional experiments by systematically varying each of these components. In addition, we also investigated BLIP (Li et al., 2022), a different method for vision-language pre-training. Due to computational resource constraints, we repeated these experiments only for the Clipart and Sketch domains, where increasing domain diversity yielded the most significant robustness gains.

**Architecture**    For the architecture experiments, we trained two CLIP configurations with different image encoder architectures. The first configuration used a Swin-T (Liu et al., 2021), the same text encoder as in our ResNet-50 experiments, and an embedding dimension of 512. The second configuration used a ViT-S-32 (Touvron et al., 2021), a slightly smaller text

encoder with a width of 384, only six attention heads, and also a smaller embedding dimension of 384. In addition, for ViT-S-32, we used slightly adjusted AdamW hyperparameters, i.e., $\beta_2 = 0.98$ and $\epsilon = 10^{-6}$. All other hyperparameters were consistent with the ResNet-50-based experiments (see Appendix A.6).

Table 7 and Figure 9 confirm that for transformer-based vision encoders, robustness also improves with increasing domain diversity, as expected. Similarly, compositional generalization can perform worse than domain generalization.

Table 7: **Results (balanced top-1 accuracy) when varying CLIP's vision encoder.** We find similar trends across these vision encoder choices.

| Vision encoder | Training data setup | Clipart | Sketch |
|---|---|---|---|
| ResNet-50 | Natural-only | 19.4 | 19.6 |
| | Leave-out-domain | 27.1 | 31.8 |
| | CG low-diversity | 18.7 | 16.4 |
| | CG high-diversity | 28.6 | 28.6 |
| ViT-S-32 | Natural-only | 12.0 | 5.9 |
| | Leave-out-domain | 15.1 | 8.2 |
| | CG low-diversity | 12.7 | 5.8 |
| | CG high-diversity | 13.8 | 8.3 |
| Swin-T | Natural-only | 17.1 | 11.3 |
| | Leave-out-domain | 20.3 | 15.2 |
| | CG low-diversity | 17.4 | 10.4 |
| | CG high-diversity | 22.3 | 14.0 |

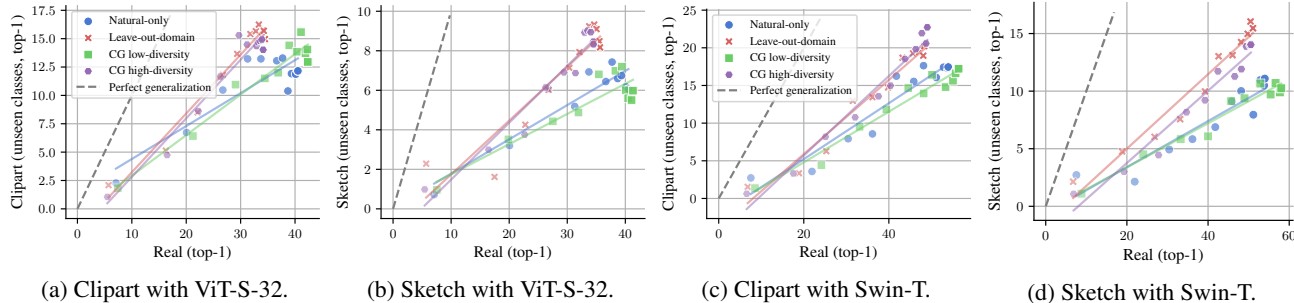

(a) Clipart with ViT-S-32.   (b) Sketch with ViT-S-32.   (c) Clipart with Swin-T.   (d) Sketch with Swin-T.

Figure 9: **Effective robustness plots for different vision encoders.** Refer to Figure 2 for the effective robustness plots for the ResNet-50 vision encoder.

**Base Dataset**   For the base dataset experiments, we trained our ResNet-50 CLIP configuration using CC3M and CC12M as the base datasets. Following Radford et al. (2021), we used a maximum learning rate of 5e-4. We also adjusted the number of warmup steps to 2000 and the batch size to 2048. All other hyperparameters were consistent with the experiments from the main text (see Appendix A.6 for the hyperparameters).

Table 8 and Figure 10 confirm that both CC3M and CC12M exhibit the same robustness trends as our ImageNet-Captions models. However, the robustness gains are smaller compared to ImageNet-Captions. This is most likely due to the reduced relative weighting of the DomainNet images with larger base dataset sizes, as well as the higher inherent diversity of CC3M and CC12M. For example, both CC3M and CC12M include some non-natural images, which may diminish the effect of further increasing domain diversity. Note that compositional generalization seems to be slightly better than domain generalization for larger datasets, which we attribute to domain contamination (see Appendix B.2 why this may benefit compositional generalization). We leave further investigation into the effect of dataset size and diversity of the base dataset for future work (see Section 7 for a discussion).

Table 8: **Results when varying the base dataset $D_0$.** We observe similar trends for CC3M and CC12M as for ImageNet-Captions. The only exception is that compositional generalization now tends to always work better than domain generalization for CC12M. We attribute this to a domain contamination of CC12M, i.e., manual inspection shows that CC12M contains a lot of Sketch and Clipart images.

| Base dataset $D_0$ | Training data setup | Clipart | Sketch |
|---|---|---|---|
| | Natural-only | 19.4 | 19.6 |
| ImageNet-Captions | Leave-out-domain | 27.1 | 31.8 |
| | CG low-diversity | 18.7 | 16.4 |
| | CG high-diversity | 28.6 | 28.6 |
| | Natural-only | 32.2 | 31.5 |
| CC3M | Leave-out-domain | 31.6 | 35.1 |
| | CG low-diversity | 28.4 | 30.5 |
| | CG high-diversity | 35.5 | 33.6 |
| | Natural-only | 39.6 | 48.3 |
| CC12M | Leave-out-domain | 46.1 | 51.2 |
| | CG low-diversity | 38.5 | 41.7 |
| | CG high-diversity | 48.7 | 51.7 |

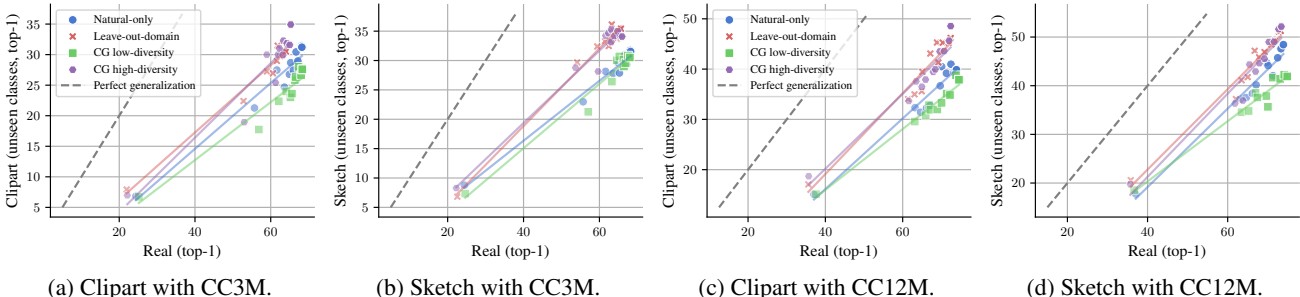

| (a) Clipart with CC3M. | (b) Sketch with CC3M. | (c) Clipart with CC12M. | (d) Sketch with CC12M. |

Figure 10: **Effective robustness plots for the different base datasets.** Refer to Figure 2 for the effective robustness plots for the ImageNet-Captions dataset.

**Contrastive Loss** For the loss experiments, we trained our ResNet-50 CLIP configuration but replaced the original CLIP loss with the SigLIP loss (Zhai et al., 2023).

Table 9 and Figure 11 confirm that the results for models trained with SigLIP are consistent with our observations from the experiments in the main text.

**Beyond Contrastive Losses** Finally, we conducted an experiment using BLIP (Li et al., 2022) instead of CLIP. Similar to CLIP, BLIP is also a vision-language pre-training method that uses a contrastive image-text loss, but additionally includes an image-text matching as well as a language modeling loss. We used the training pipeline provided by LAVIS (https://github.com/salesforce/LAVIS, License: BSD 3-Clause, Li et al. (2023)) to train BLIP in its base configuration, which uses a ViT-B-16 (Dosovitskiy et al., 2021) pre-trained on ImageNet as the image encoder and a text encoder initialized from BERT$_{base}$ (Devlin et al., 2019).

Table 10 and Figure 12 show that the results for BLIP follow the same trends as our CLIP models, except for the overall higher performance due to the pre-trained image and text encoders.

### B.2. Challenges of Compositional Generalization

In Section 5, we found that compositional generalization settings can suffer from an exposure to a subset of classes during training that are later queried in evaluation. This can make compositional generalization fail but can be alleviated by using domain samples from classes that do not overlap with the class distribution which is queried during evaluation. Table 2 shows

Table 9: **Results when using a different loss function (SigLIP (Zhai et al., 2023)).** The choice of loss function does not change the trends observed for CLIP's original contrastive loss.

| Loss function | Training data setup | Clipart | Sketch |
|---|---|---|---|
| CLIP | Natural-only | 19.4 | 19.6 |
| | Leave-out-domain | 27.1 | 31.8 |
| | CG low-diversity | 18.7 | 16.4 |
| | CG high-diversity | 28.6 | 28.6 |
| SigLIP | Natural-only | 20.4 | 19.7 |
| | Leave-out-domain | 28.1 | 26.1 |
| | CG low-diversity | 19.2 | 16.4 |
| | CG high-diversity | 25.7 | 27.5 |

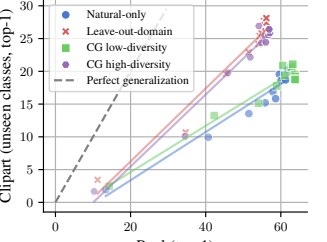 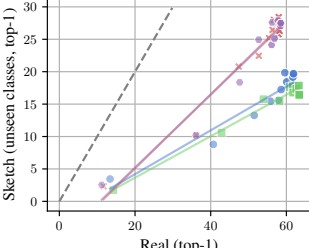

Figure 11: **Effective robustness plots for SigLIP.** Refer to Figure 2 for effective robustness plots when using CLIP's original contrastive loss.

that replacing *all* DomainNet sketches with sketches that do not overlap with DomainNet's classes (see Appendix A.3), using ImageNet-Sketch, improves compositional generalization performance from 28.1% to 36.9%.

In practice, however, enforcing little to no class overlap may not always be feasible; particularly in zero-shot settings where CLIP is applied to data and/or tasks that are unknown at training time. Therefore, we investigated the severity of this in more detail. To do this, we partitioned the set of classes $C = C_1 \cup C_2 \cup C_3$ in our test domain $D_i$ into three disjoint subsets (see Figure 13a):

- $C_1$: Classes that are seen during training and are queried during evaluation.

- $C_2$: Classes that are *not* seen during training and are queried during evaluation. Note that only classes from $C_2$ are contained in the test set $D_i^{C_2}$.

- $C_3$: Classes that are seen during training and are *not* queried during evaluation.

This partitioning allowed us to systematically assess the impact of class overlap on compositional generalization by constructing training sets with varying mixtures of classes from $C_1$ and $C_3$.

For this experiment, we selected Sketch as our test domain.[5] The subsets $C_1$ and $C_2$ were defined as described in Section 3, meaning that $C_1 \cup C_2$ represents the class distribution queried during evaluation (i.e., all DomainNet classes). For $C_3$, we used classes from ImageNet-Sketch (Wang et al., 2019) that do not overlap with the classes from DomainNet (see Appendix A.3 for more details). Note that throughout all these experiments, the class distribution in the other domains $D_{j \neq i}$ remained unchanged—that is, classes from $C_1$ and $C_2$ were included in training, while no classes from $C_3$ were introduced.

As shown in Table 2 and Figure 13b (rightmost bar), training solely on samples from $C_3$ significantly improves compositional generalization performance compared to not seeing any test domain samples from $D_i$ (zero line) or only seeing samples from classes $C_1$ of $D_i$ (leftmost bar). To further investigate this, we trained CLIP models using samples from $C_1 \cup C_3$ of the test domain $D_i$ and gradually reduced the number of classes from $C_1$, while keeping all other factors fixed. Figure 13b shows

---

[5]We chose Sketch due to the availability of a large class distribution from ImageNet-Sketch.

Table 10: **Results when using BLIP (Li et al., 2022).** Using BLIP as a pre-training method does not change the trends observed for standard CLIP. Note that the overall increased performance of BLIP is due to their usage of pre-trained image and text encoders.

| Training regime | Training data setup | Clipart | Sketch |
|---|---|---|---|
| | Natural-only | 19.4 | 19.6 |
| CLIP | Leave-out-domain | 27.1 | 31.8 |
| | CG low-diversity | 18.7 | 16.4 |
| | CG high-diversity | 28.6 | 28.6 |
| | Natural-only | 37.1 | 35.7 |
| BLIP | Leave-out-domain | 61.8 | 58.6 |
| | CG low-diversity | 35.0 | 41.6 |
| | CG high-diversity | 55.6 | 61.2 |

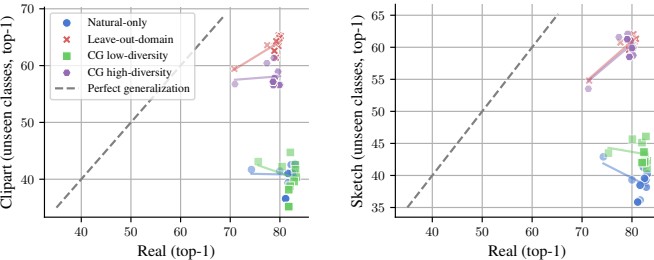

Figure 12: **Effective robustness plots for BLIP.** Refer to Figure 2 for effective robustness plots when using standard CLIP.

that reducing class overlap (moving from left to right)—i.e., decreasing the number of $C_1$ classes seen during training—consistently improves compositional generalization performance. This reaffirms the detrimental effect of class overlap, as observed in Table 2, but also highlights that increasing class diversity can help mitigate its impact on compositional generalization performance.

**Supervised classifiers** We also investigated to what extent supervised classifiers are vulnerable to this bias. Table 11 confirms that supervised classifiers are also susceptible to it.

### B.3. Closing The Generalization Gap

Table 6 clearly shows that there is a significant performance gap between the CG high-diversity with and without including domain-specific samples of our test classes $C_2$, even for domains like Clipart and Sketch where generalization seems to work reasonably well. We conducted an interpolation experiment between the two settings with and without such samples to better understand how many of these samples are actually required to close this "generalization gap". To further investigate how the generalization capability of a model impacts the required number of samples, we also performed the same experiment for the CG low-diversity setting.

For a given test domain $D_i$, we consider the number of training samples of our test set $D_i^{C_2}$ to be 100% and then trained additional CLIP models in which we successively added 5%, 10%, 15%, 20%, 40%, 60%, and 80% of training samples from $D_i^{C_2}$ to the training data. Note that we ensured that the overall dataset size remained unchanged.

Figure 14 shows that performance on the classes seems to follow a roughly linear relationship with the number of samples allowed for all domains, except for the Quickdraw domain. For Quickdraw, the relationship seems to be log-linear instead, which may be due to the fact that CLIP does not generalize at all for Quickdraw. This observation also relates to the findings of Udandarao et al. (2024), who predict that a linear increase in samples leads only to a log-linear increase in zero-shot performance. Across all domains and settings, we observed that to achieve the maximally possible performance, 100% of the training samples from $D_i^{C_2}$ are required.

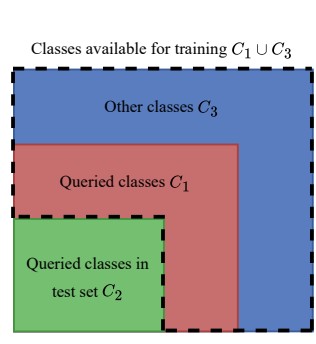

(a) Classes from the test domain.

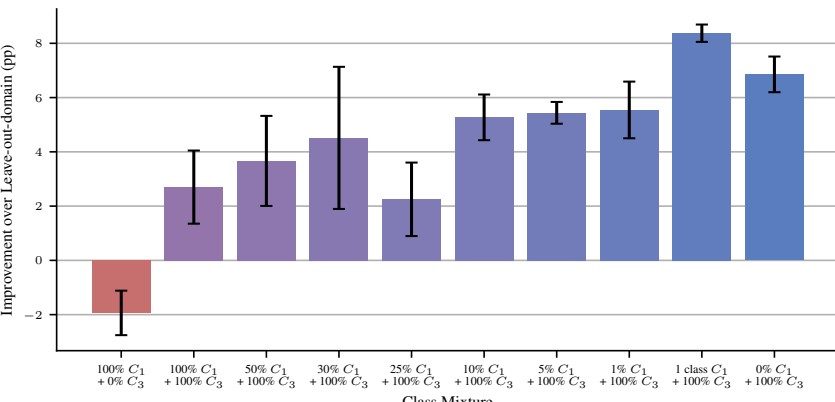

(b) Effect of different mixtures of the test domain's classes $C_1 \cup C_3$ seen during training.

Figure 13: **Severity of class overlap on compositional generalization. a**: We partitioned the classes $C$ in the test domain $D_i$ into three disjoint subsets: $C = C_1 \cup C_2 \cup C_3$. During evaluation, both $C_1$ and $C_2$ are included in the query set, allowing the model to predict any class from these subsets. However, the actual test set, $D_i^{C_2}$, only contains samples from $C_2$. In contrast, the classes in $C_3$ are excluded from both the query and test sets. This partitioning allows us to investigate the severity of class overlap for compositional generalization performance. In particular, we found that including classes of $C_1$ led to a significant drop in balanced top-1 accuracy for compositional generalization—even performing worse than domain generalization (Figure 2). On the other hand, replacing the class samples from $C_1$ with the ones of $C_3$ (which are not part of the query set) resulted in a significant improvement in compositional generalization (Table 2). **b**: To investigate the severity, we varied the mixture of classes from $C_1$ and $C_3$. We find that reducing classes from $C_1$ (moving from left to right), while keeping all other factors fixed, steadily improves compositional generalization performance (with the exceptions for the class mixtures 25% $C_1$ + 100% $C_3$ and 1 class $C_1$ + 100% $C_3$).

Table 11: **Supervised classifiers are also susceptible to the seen class bias.** Supervised classifiers' compositional generalization also deteriorates due to a partial test domain overlap and replacing them with samples from non-overlapping classes significantly improves compositional generalization.

| Training data setup (Figure 1) | Sketch |
|---|---|
| Leave-out-domain | 27.4 |
| CG high-diversity | 22.2 |
|     w/ sketches of non-queried classes only | 30.1 (+7.9) |

## B.4. Role of Language Supervision

Previous studies comparing the robustness of CLIP models and supervised classifiers either examined models trained on different datasets or focused on low-diversity datasets consisting mostly of natural images, such as ImageNet-Captions (Fang et al., 2022). Since both ImageNet-Captions and DomainNet provide class labels, we investigated how our domain mixtures affect the robustness of supervised classifiers and compared the results to CLIP models.

Table 12 and Figure 15 show the results of our experiments on supervised classifiers. We found that CLIP models consistently exhibit slightly higher robustness than their supervised counterparts, which may be due to the richness of captions (Xue et al., 2024; Wen et al., 2024). Interestingly, CLIP's advantages are more pronounced in compositional generalization settings (CG low/high-diversity). However, in the domain generalization setting (Leave-out-domain), supervised classifiers can sometimes achieve generalization comparable to CLIP.

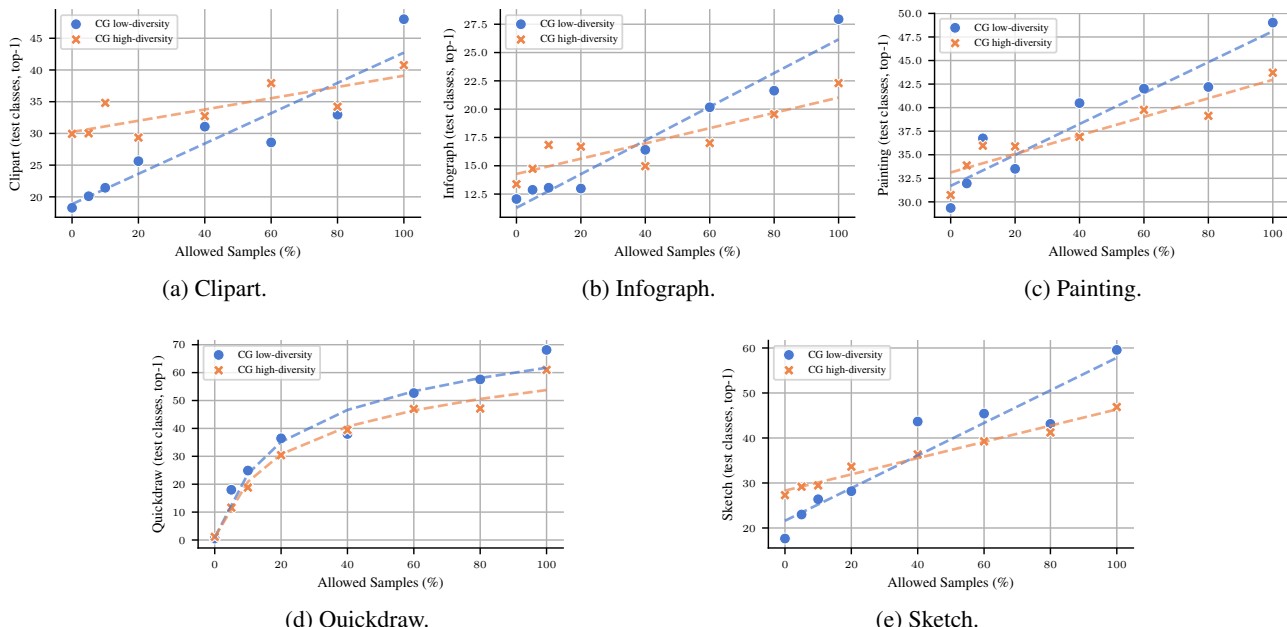

(a) Clipart.  (b) Infograph.  (c) Painting.

(d) Quickdraw.  (e) Sketch.

Figure 14: **Interpolation between CG settings with and without domain-specific training samples of the test classes** $C_2$. For each interpolation, we fitted both a linear and a log-linear regression model and visualized the fit with a lower mean squared error (MSE). Balanced top-1 accuracy on the test classes appears to follow a roughly linear relationship with the number of test samples included, except for Quickdraw, which shows a strong log-linear relationship.

Table 12: **Performance comparison across supervised experiments.** The robustness of supervised models (as measured by balanced top-1 accuracy) also increases with domain diversity. However, supervised classifiers generalize slightly worse than CLIP models (Figures 4 and 15).

|  | Clipart | Infograph | Painting | Quickdraw | Sketch |
|---|---|---|---|---|---|
| Natural-only | $17.6 \pm 1.6$ | $10.9 \pm 0.8$ | $32.6 \pm 1.6$ | $0.7 \pm 0.1$ | $15.0 \pm 1.1$ |
| Leave-out-domain | $30.8 \pm 2.1$ | $14.5 \pm 1.2$ | $35.7 \pm 1.2$ | $10.1 \pm 0.5$ | $27.4 \pm 2.1$ |
| CG low-diversity | $13.2 \pm 0.6$ | $9.7 \pm 0.5$ | $25.1 \pm 0.9$ | $0.0 \pm 0.0$ | $12.0 \pm 1.1$ |
| CG high-diversity | $25.0 \pm 1.6$ | $12.2 \pm 0.7$ | $28.6 \pm 0.4$ | $0.7 \pm 0.4$ | $22.2 \pm 0.7$ |

## C. Additional Technical Details and Results for Section 6

### C.1. Technical Details on Sparse Autoencoders

Following (Bricken et al., 2023; Huben et al., 2024), we used a Sparse Autoencoder (SAE) to extract interpretable features from CLIP's visual embeddings $\mathbf{a} \in \mathbb{R}^p$. The SAE is defined as follows:

$$\text{SAE}(\mathbf{a}) := (g \circ \phi \circ f)(\mathbf{a}), \tag{4}$$

where $\phi$ is a ReLU non-linearity, and $f$ and $g$ are linear encoder with weights $\mathbf{W}_f \in \mathbb{R}^{p \times h}$ or decoder with weights $\mathbf{W}_g \in \mathbb{R}^{h \times p}$, respectively. We trained the SAE with an $L_2$ reconstruction loss and $L_1$ sparsity regularization:

$$\mathcal{L}(\mathbf{a}) = \|\mathbf{a} - (g \circ \phi \circ f)(\mathbf{a})\|_2^2 + \lambda \|(\phi \circ f)(\mathbf{a})\|_1, \tag{5}$$

where $\lambda$ governs the sparsity regularization strength.

We trained SAE's on the activations of our CC12M CLIP models[6] (see Appendix B.1). We used CC12M and the complete

---

[6] We also tried the CLIP models that were trained with ImageNet-Captions as base dataset but found that the SAE extracted poorly interpretable features.

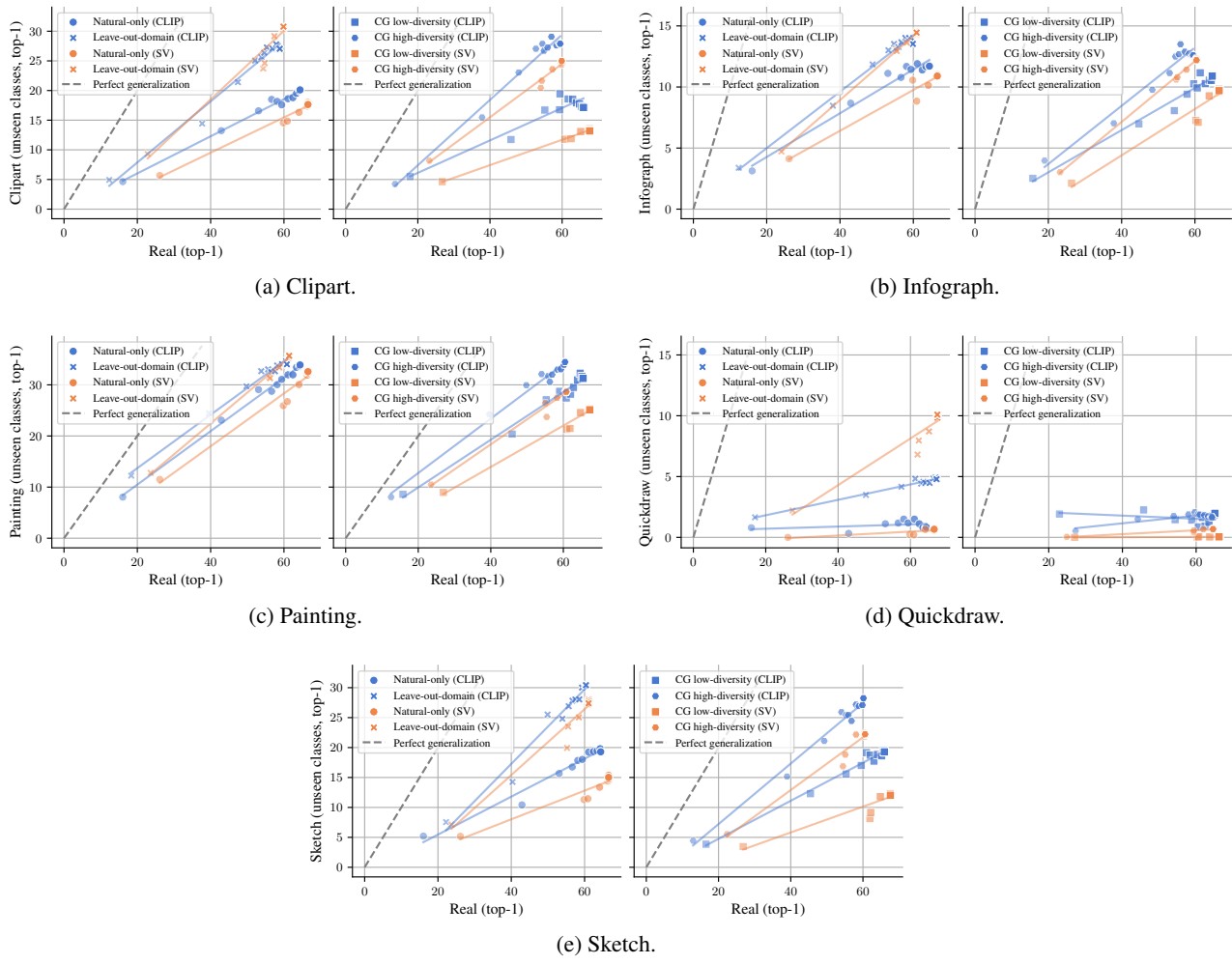

Figure 15: **Effective robustness plots for CLIP vs. supervised classifiers.** While the general trends are very similar between CLIP models and supervised classifiers, CLIP models typically generalizes better than the supervised classifiers.

DomainNet training set to train the SAE's to identify interpretable features. The hidden dimension $h$ was set to 4096, i.e., 4x the embedding dimension of the CLIP model's output dimensionality. We trained the SAE for 200 epochs using a batch size of 4096. The regularization strength hyperparameter $\lambda$ was set to 1e-4. To alleviate the dying neuron problem, dead neurons were resampled every 500,000 training steps. Our implementation is based on the code published by Rao et al. (2024) (`https://github.com/neuroexplicit-saar/discover-then-name`, License: MIT).

We trained one SAE for our natural-only baseline, two SAE's for the leave-out-domain setting (with Clipart and Sketch as the test domain), and another two SAEs for the CG high-diversity setting (also with Clipart and Sketch as test domain). For each model, we analyzed the extent to which the top-k most activating SAE features are shared between the test domain (Clipart or Sketch) and all other domains (not just the training domains).

To identify the top-k SAE features, we computed the SAE hidden representations (after applying the non-linearity $\phi$) for each sample of each class and domain. We then selected the top-k SAE features (with $k \in \{5, 10, 15, 20\}$) per class-domain pair based on how frequently they ranked among the top-20 most activating features, i.e., largest activation magnitudes (see Algorithm 1).

To measure feature sharing, we then calculated the percentage overlap between the top-k SAE features of the test domain and those of each of the other domains. To yield a single overlap score per model, we averaged across classes $C_2$, these pairs of domains, and the four values of $k$ (see Algorithm 2). Finally, we calculated the increase of the overlap score of the corresponding leave-out-domain and CG high-diversity models over the natural-only baseline model.

---

**Algorithm 1** get_topk_features

---

**input** domain of interest $D_i$, class of interest $c$, number of features to consider $k$

1: hist $\leftarrow \mathbf{0} \in \mathbb{R}^h$                                                                     // $h$ is the number of concepts
2: **for** each image $I \in D_i^c$ **do**
3:     a $\leftarrow f_{\text{CLIP}}(I)$                                                                     // CLIP image activation
4:     $w \leftarrow \phi(f_{\text{SAE}}(a))$                                                                     // SAE concept activations
5:     **for** each index $i \in \text{topk}(w, 20)$ **do**
6:         hist$[i] \leftarrow$ hist$[i] + 1$                                                                     // count top-20 activating concepts
7:     **end for**
8: **end for**
**output** topk(hist, $k$)                                                                     // return top-k most frequent features

---

**Algorithm 2** measure_feature_sharing

---

**input** test domain $D_i$

1: $S \leftarrow \emptyset$                                                                     // initialize empty set of scores
2: **for** each $k \in \{5, 10, 15, 20\}$ **do**
3:     **for** each class $c \in C_2$ **do**
4:         $F_i \leftarrow$ get_topk_features$(D_i, c, k)$                                                                     // top-k features of test domain $D_i$
5:         **for** each other domain $D_j \neq D_i$ **do**
6:             $F_j \leftarrow$ get_topk_features$(D_j, c, k)$                                                                     // top-k features of domain $D_j$
7:             $s \leftarrow \frac{|F_i \cap F_j|}{k}$                                                                     // percentage overlap of $F_i$ and $F_j$
8:             $S \leftarrow S \cup \{s\}$
9:         **end for**
10:     **end for**
11: **end for**
**output** $\frac{1}{|S|} \sum_{s \in S} s$                                                                     // overlap averaged over domains $D_j$, classes $c$, and $k$

---

## C.2. Technical Details on Center Kernel Alignment and Additional Results

Let $\mathbf{X}^{D_1} \in \mathbb{R}^{C \times p}$ and $\mathbf{Y}^{D_2} \in \mathbb{R}^{C \times p}$ contain the $C$ mean visual embeddings of each class for the domains $D_1$ or $D_2$, respectively. Then, we compute the Gram matrices/kernels $\mathbf{K} = \mathbf{X}^{D_1}(\mathbf{X}^{D_1})^T$ and $\mathbf{L} = \mathbf{Y}^{D_2}(\mathbf{Y}^{D_2})^T$ that contain the pairwise similarities of each pair of mean class embeddings. Note that we used a linear kernel here, as commonly done in the representational similarity literature. Alternatively, we also tried a non-linear kernel (i.e., RBF kernel) with similar results (see Figure 17). Center kernel alignment is defined by Kornblith et al. (2019) as follows:

$$\text{CKA}(\mathbf{K}, \mathbf{L}) = \frac{\text{HSIC}(\mathbf{K}, \mathbf{L})}{\sqrt{\text{HSIC}(\mathbf{K}, \mathbf{K})\text{HSIC}(\mathbf{L}, \mathbf{L})}}, \tag{6}$$

where we used the used the unbiased Hilbert-Schmidt Independence Criterion (HSIC) estimator (Song et al., 2012) following Nguyen et al. (2021), defined as follows:

$$\text{HSIC}(\mathbf{K}, \mathbf{L}) = \frac{1}{C(C-3)}\Big(tr(\tilde{\mathbf{K}}\tilde{\mathbf{L}}) + \frac{\mathbf{1}^T\tilde{\mathbf{K}}\mathbf{1}\mathbf{1}^T\tilde{\mathbf{L}}\mathbf{1}}{(C-1)(C-2)} - \frac{2}{C-2}\mathbf{1}^T\tilde{\mathbf{K}}\tilde{\mathbf{L}}\mathbf{1}\Big), \tag{7}$$

where we set the diagonal elements of $\mathbf{K}$ and $\mathbf{L}$ to zero in $\tilde{\mathbf{K}}$ or $\tilde{\mathbf{L}}$, respectively.

We computed center kernel alignment between all pairs of domains for each class. Thereby, we can assess the representational similarity for each class across the domains. For visualization, for each domain, we averaged over all classes to obtain a representational similarity estimate. In the main text, we further averaged together the estimates for the non-Quickdraw domains ('Others'). The estimates for each individual domain is shown in Figures 16 and 17.

ADDITIONAL RESULTS

Figures 16 and 17 show results for all classes $C$, the classes seen during training $C_1$, and the unseen classes $C_2$ for the CLIP model solely trained on domain-invariant captions from Figure 5b and Table 4 (second row). Interestingly, we find that the

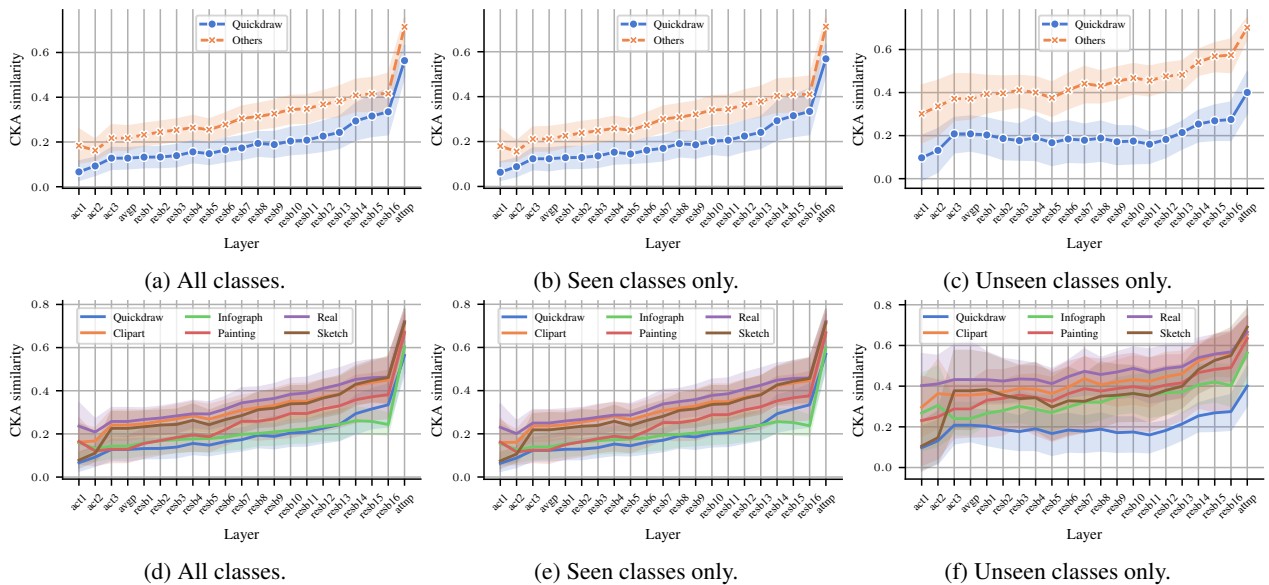

Figure 16: **Linear center kernel alignment (CKA) similarity.** Quickdraw has the lowest representational similarity across domains. This particularly emphasized for the unseen classes $C_2$ (c, f). We used the CLIP model trained on only domain-invariant captions from Figure 5b and Table 4 (second row). Figure 18 provides results for other CLIP models.

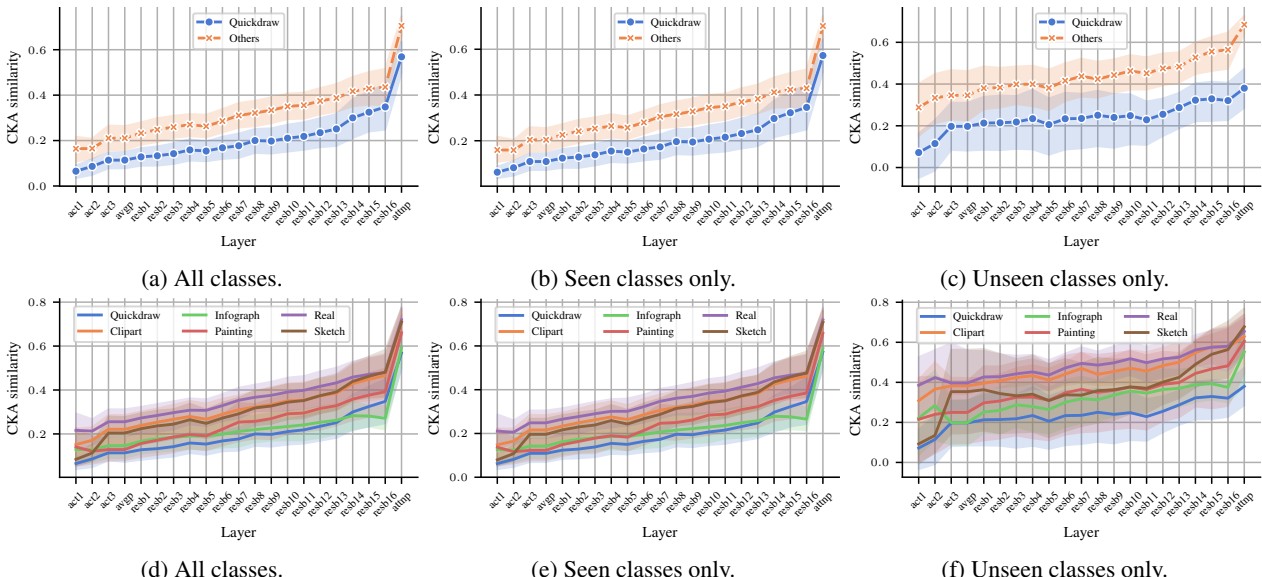

Figure 17: **Non-linear center kernel alignment (CKA) similarity.** The results follow the same pattern as the linear CKA (Figures 6a and 16): Quickdraw is the most representational dissimilar domain. We used the CLIP model trained on only domain-invariant captions from Figure 5b and Table 4 (second row).

representational gap widens for the unseen classes $C_2$. Figure 18 shows the representational similarity for different CLIP models from Figure 2 for the unseen classes $C_2$. Similar to the above, the CLIP model evaluated on the Quickdraw domain exhibits low representational similarity to models from other domains. In contrast, the CLIP models with the test domains Clipart, Infograph, Painting, and Sketch show higher or more comparable representational similarities.

### C.3. Technical Details on the Circuit Similarity Analysis and Additional Results

Attributing the causal effects of model components is a key goal of (mechanistic) interpretability research (Meng et al., 2022; Marks et al., 2025; Mueller et al., 2024). We drew inspiration from them to analyze the level of sharing of the most

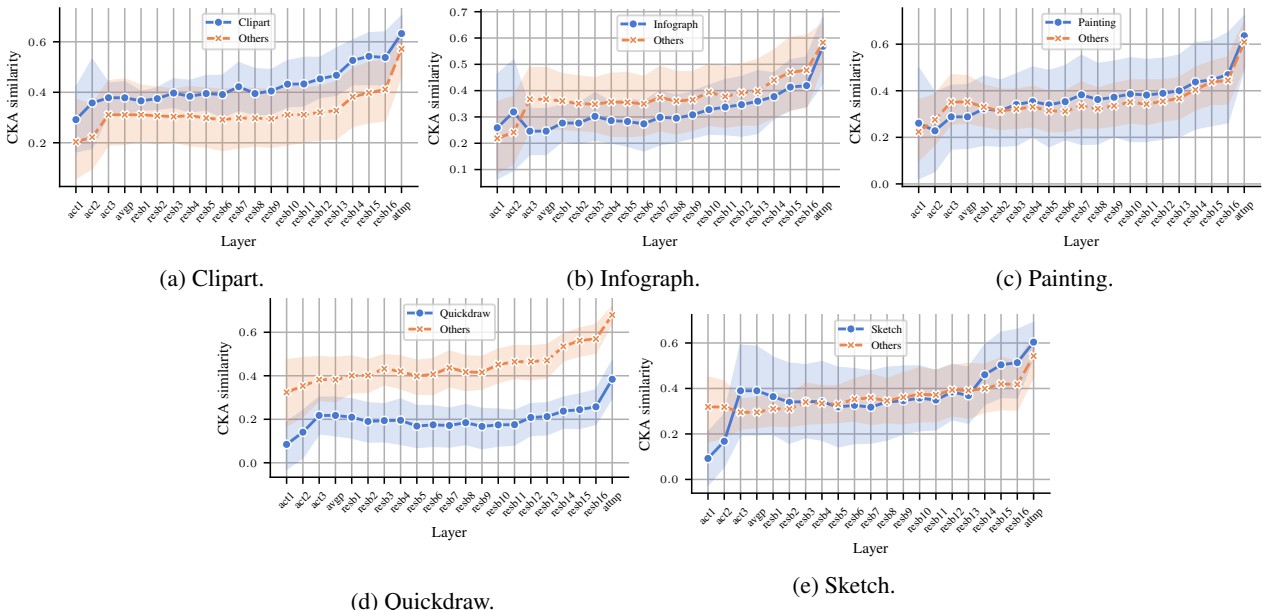

(a) Clipart.        (b) Infograph.        (c) Painting.

(d) Quickdraw.        (e) Sketch.

Figure 18: **Linear center kernel alignment (CKA) similarity for different CLIP models.** We used the CLIP models from the CG high-diversity setting in Figure 2 and report representational similarities on the unseen classes $C_2$. The patterns for all classes $C$ and for the seen classes $C_1$ are qualitatively similar. Among the domains, Clipart, Painting, and Sketch exhibit comparable levels of representational similarity, whereas Infograph and particularly Quickdraw show lower similarities.

important CLIP's model components, i.e., the axis-aligned neurons in its vision encoder, across domains. To do this, we first must attribute the components' importance via the indirect effect (Pearl, 2001).

Let $m_{:l}(I_{\text{clean}}^d) = \mathbf{a}_{\text{clean}}^{l,n} \in \mathbf{p}$ be a $p$-dimensional neuron $n$ in the $l$-th layer of model $m$ for input image $I_{\text{clean}}$. Further, we define $\mathbf{a}_{\text{patch}}^{l,n} \in \mathbf{p}$ as a corrupted baseline. The computation of the indirect effect is defined as follows:

$$\text{IE} = m_{L,c}(I_{\text{clean}}^d | \text{do}(\mathbf{a}^l = \mathbf{a}_{\text{patch}}^{l,n})) - m_{L,c}(\mathbf{a}_{\text{clean}}^{l,n}), \tag{8}$$

where $m_{L,c}$ is the output logit for class $c$ of the last layer $L$ (note that we can obtain this logit through the dynamic zero-shot weights that can be generated by CLIP's text encoder, see Appendix A.5) and $\text{do}(\mathbf{a}^l = \mathbf{a}_{\text{patch}}^{l,n})$ denotes the do-operator (Pearl, 2009) that intervenes on the computation of the CLIP model by setting the activations $\mathbf{a}^l$ to $\mathbf{a}_{\text{patch}}^{l,n}$.

Since there are lot of neurons in CLIP's vision encoder, we sped up the computation through a linear approximation using integrated gradients (Sundararajan et al., 2017), following Marks et al. (2025); Hanna et al. (2024):

$$\hat{\text{IE}}_{\text{ig}} = \left( \sum_\alpha \nabla_{\mathbf{a}^{l,n}} m_{l:}(\alpha \mathbf{a}_{\text{clean}}^{l,n} + (1-\alpha)\mathbf{a}_{\text{patch}}^{l,n}) \right) (\mathbf{a}_{\text{patch}}^{l,n} - \mathbf{a}_{\text{clean}}^{l,n}), \tag{9}$$

where $\alpha \in \{0, \frac{1}{N}, \cdots, \frac{N-1}{N}\}$ and we set $\mathbf{a}_{\text{patch}}^{l,n}$ to $\mathbf{0}$. We adapted the codebase from Marks et al. (2025) (https://github.com/saprmarks/feature-circuits, License: MIT)—using nnsight (Fiotto-Kaufman et al., 2025, https://github.com/ndif-team/nnsight, License: MIT) internally—for our implementation.

DISCOVERY OF CIRCUITS

We use above computation for the indirect effect to find the $k$ most important neurons and $k'$ most important preceding neurons of each of those neurons for *each* class of *each* domain. This approach is outlined below in more detail:

1. **Identify the $k$ most important neurons**: We directly apply Equation 9. We used $N = 10$ steps for the linear approximation and only retained the 10% most important neurons per layer. Note that these neurons represent the nodes of the graph representing the circuit.

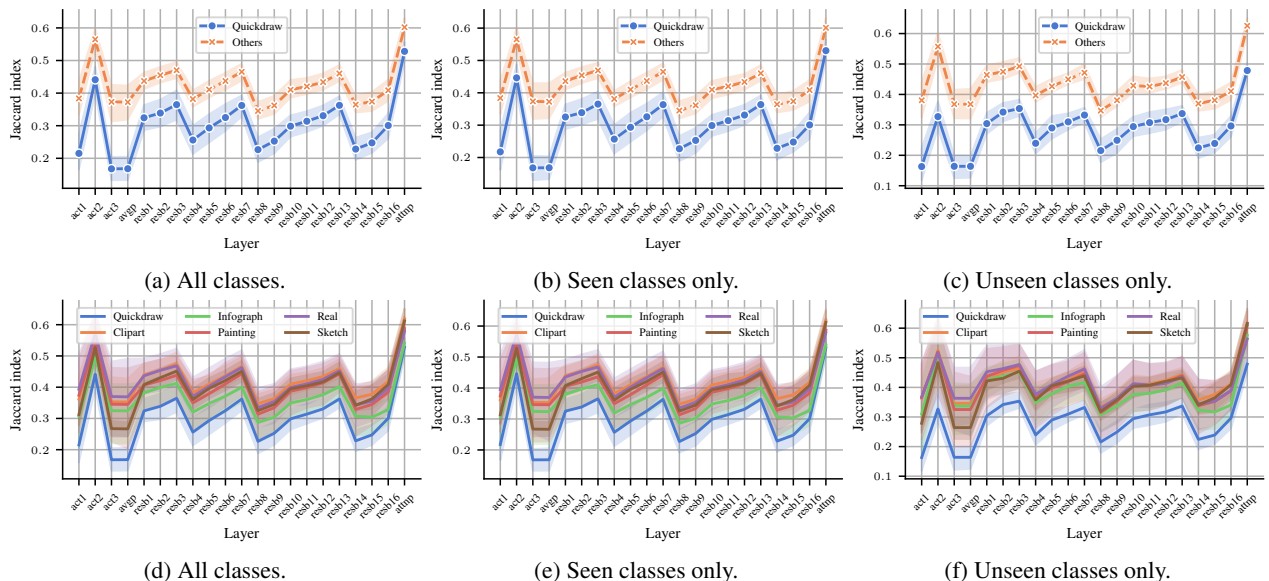

(a) All classes.          (b) Seen classes only.          (c) Unseen classes only.

(d) All classes.          (e) Seen classes only.          (f) Unseen classes only.

Figure 19: **Amount of shared neurons.** The Quickdraw domains shares the least neurons. This is especially apparent for the unseen classes $C_2$ (c, f). Results are for the CLIP model trained on only domain-invariant captions from Figure 5b and Table 4 (second row). Figure 20 provides results for other CLIP models.

Table 13: **Weisfeiler-Lehman similarities.** Higher similarities mean higher circuit (graph) similarity. We used the CLIP model trained on only domain-invariant captions from Figure 5b and Table 4 (second row); results for other CLIP models are provided in Table 14. The Quickdraw domain exhibits the least degree of similarity, supporting our hypothesis that sharing of circuits is critical for generalization.

|  | Clipart | Infograph | Painting | Quickdraw | Real | Sketch |
|---|---|---|---|---|---|---|
| All classes | 0.281 | 0.246 | 0.266 | 0.218 | 0.273 | 0.273 |
| Seen classes only | 0.281 | 0.246 | 0.265 | 0.218 | 0.273 | 0.273 |
| Unseen classes only | 0.278 | 0.258 | 0.274 | 0.211 | 0.272 | 0.275 |

2. **Identify the $k'$ most important preceding neurons of each neuron**: We adapted Equation 9 to measure the effect of preceding neurons $n$ of layer $l < l'$ on the activations of neuron $n'$ in layer $l'$. Specifically, we replaced $m_{L,c}$ by $m_{l',n'}$ and measured the $L_2$ change caused by the clean activations $\mathbf{a}_{\text{clean}}^{l,n}$ and intervened activations $\alpha \mathbf{a}_{\text{clean}}^{l,n} + (1-\alpha)\mathbf{a}_{\text{patch}}^{l,n}$. We also set $N$ to 10. Note that this will yield us edges for the graph representing the circuit. After computing the indirect effects of all preceding neurons, we only retained the $k' = 3$ most important edges.

MEASURING CIRCUIT SIMILARITY

We compared the circuit similarity of pairs of circuits (graphs) resorting to classical graph similarity measures. Similar to the representational analysis, we computed the graph similarity measures between all pairs of domains for each class and averaged over all classes to obtain the final circuit similarity estimates. In the main text, we further averaged together the estimates of the non-Quickdraw domains ('Others'). The estimates of each individual domain are shown in Figures 19d to 19f and Table 13.

**Node-level similarity**    We computed the Jaccard index for the nodes $N_{C_{\text{domain } D_1, \text{class } c}}$, $N_{C_{\text{domain } D_2, \text{class } c}}$, as follows:

$$\frac{|N_{C_{\text{domain } D_1, \text{class c}}} \cap N_{C_{\text{domain } D_2, \text{class c}}}|}{|N_{C_{\text{domain } D_1, \text{class c}}} \cup N_{C_{\text{domain } D_2, \text{class c}}}|} \quad . \tag{10}$$

Intuitively, this quantifies the degree of overlap between the most contributing model components for class $c$ across domains $D_1$ and $D_2$.

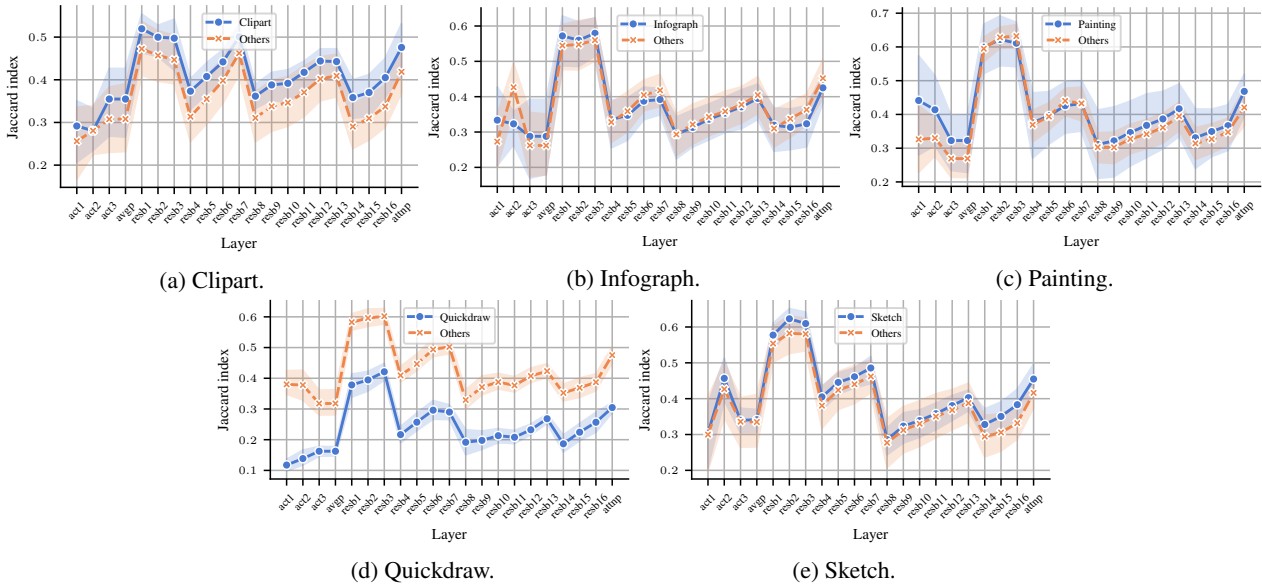

Figure 20: **Amount of shared neurons for different CLIP models.** We used the CLIP models from the CG high-diversity setting in Figure 2 and report results on the unseen classes $C_2$. While Clipart, Infograph, Painting, and Sketch exhibit similar levels of neuron sharing, Quickdraw shows significantly less sharing.

Table 14: **Weisfeiler-Lehman similarities for different CLIP models.** Higher similarities mean higher circuit (graph) similarity. We used the CLIP models in the CG high-diversity setting from Figure 2. Results are for the unseen classes $C_2$. The CLIP models with the test domains Clipart, Painting, and Sketch have a higher degree of similarity than Infograph and Quickdraw.

| Test domain | Test domain vs. Others | Others vs. Others |
|-------------|:----------------------:|:-----------------:|
| Clipart     | 0.274                  | 0.243             |
| Infograph   | 0.246                  | 0.250             |
| Painting    | 0.263                  | 0.248             |
| Quickdraw   | 0.176                  | 0.272             |
| Sketch      | 0.259                  | 0.242             |

**Graph-level (structural) similarity**   Above simple node overlap cannot capture more complex structural and hierarchical similarities between circuits. Thus, we used the normalized Weisfeiler-Lehman subtree graph kernel (Shervashidze et al., 2011). The Weisfeiler-Lehman graph kernel is popular graph kernel choice since (1) it can handle labeled, directed graphs of different sizes, (2) it is expressive, and (3) scales well to large graphs. The main idea of the Weisfeiler-Lehman kernel is the following procedure, given two labeled graphs, $G_1$ and $G_2$:

1. **Multiset labeling and sorting**: For each node $n \in G_1$, create a multiset (unordered set with duplicates allowed) consisting of the node's $n$ current label and the sorted labels of its neighbors. Repeat this step for each node $n' \in G_2$.

2. **Label compression**: Assign a unique new label to each of these multisets using a hash function.

3. **Counting occurrences**: Count the occurrences of each compressed label to obtain the feature vectors $\phi_h(G_1)$, $\phi_h(G_2)$.

4. **Relabeling**: Replace the current node labels with the newly compressed labels.

We can repeat this procedure for $h$ iterations and compute the similarity of graphs via:

$$K(G_1, G_2) = \langle \phi(G_1),\ \phi(G_2) \rangle = \sum_h \phi_h(G_1) \cdot \phi_h(G_2) \quad . \tag{11}$$

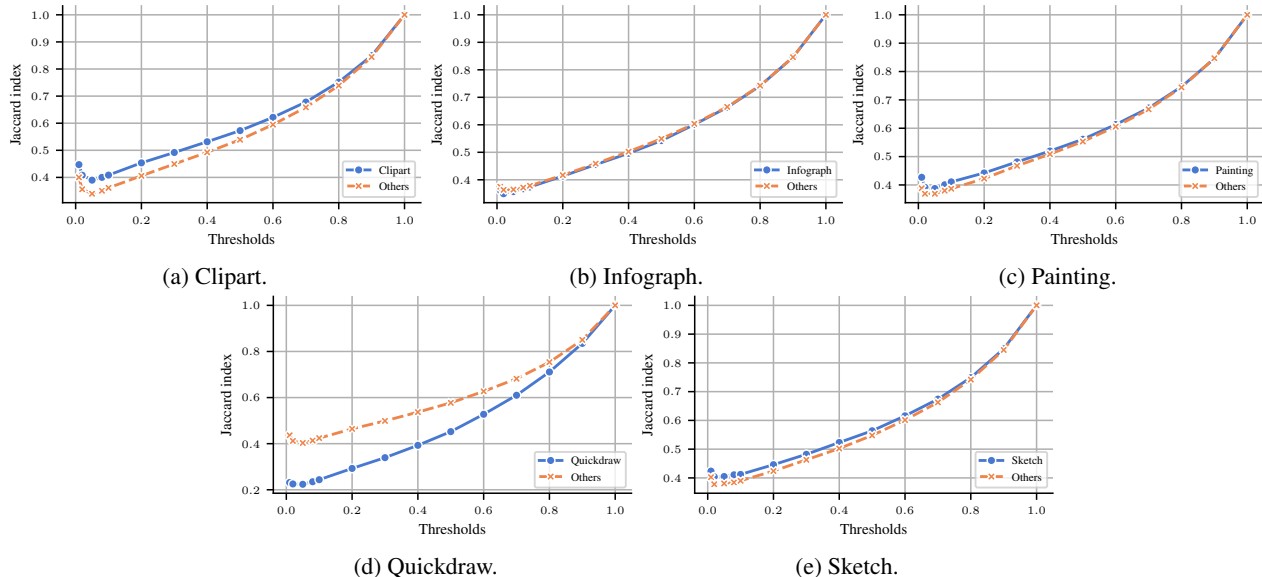

Figure 21: **Amount of shared neurons for different CLIP models over various neuron thresholds.** We used the CLIP models from the CG high-diversity setting in Figure 2 and report results on the unseen classes $C_2$, averaged across all layers and evaluated over multiple thresholds. The patterns observed in Figure 20 remain consistent across these thresholds.

Finally, we normalize the kernel to obtain a similarity score between 0 and 1:

$$\tilde{K}(G_1, G_2) = \frac{K(G_1, G_2)}{\sqrt{K(G_1, G_1) \cdot K(G_2, G_2)}} \quad .\tag{12}$$

For our analysis, we used $h = 3$ iterations. Our implementation is based on the publicly available code from `https://github.com/emanuele/jstsp2015`, License: MIT.

**False positives** In certain cases, the Weisfeiler-Lehman kernel can incorrectly identify non-isomorphic graphs as isomorphic. However, several factors reduce the chance of such false positives in our setting. Specifically, we used $h = 3$, which yields a more expressive graph kernel. In addition, node names in the circuits encode global topological information (i.e., layer and neuron indices), which further reduces the chance of false positives. Finally, we verified that the circuits were already distinguishable at the node level (i.e., at the 0-th iteration of the Weisfeiler-Lehman kernel).[7]

### ADDITIONAL RESULTS

Figure 19 and Table 13 show results for all classes $C$, the classes seen during training $C_1$, and the unseen classes $C_2$ for the CLIP model only trained on domain-invariant captions from Figure 5b and Table 4 (second row). Neurons (nodes) are less shared and the circuits are slightly less similar for the unseen classes $C_2$. Figure 20 and Table 14 extend this analysis to the different CLIP models from Figure 2 (results are on the unseen classes $C_2$). Across these models, we consistently find that domains like Clipart, Painting, and Sketch maintain similar levels of neuron and circuit sharing, whereas Quickdraw exhibits less sharing. Figure 21 shows that our observations are robust across various neuron thresholds (we chose to retain the 10% most important neurons per layer in Figures 6b, 6c, 19 and 20 and Tables 13 and 14).

---

[7]False positives can only occur if two graphs remain indistinguishable across all iterations of the Weisfeiler-Lehman kernel.

