# OpenReview forum: "When and How Does CLIP Enable Domain and Compositional Generalization?"
_ICML.cc/2025/Conference — ICML 2025 spotlightposter_

### Official Review · Reviewer_H9Z3 · 2025-02-16

**Overall Recommendation:** 4

**Summary:**

This paper investigates the domain generalization and compositional generalization capabilities of CLIP models, focusing on how the diversity of training domains affects their ability to generalize to unseen domains and unseen class-domain combinations. The authors systematically construct training distributions with controlled domain diversity and object class exposure to evaluate CLIP's performance in various settings.

**Claims And Evidence:**

yes

**Essential References Not Discussed:**

No

**Experimental Designs Or Analyses:**

The authors designed a comprehensive set of experiments to investigate CLIP's domain and compositional generalization capabilities.  The key components of their experimental design include Controlled Training Distributions, the authors systematically varied the training data by constructing four different setups: Natural-only, Leave-out-domain, CG low-diversity, and CG high-diversity.  This approach allows them to isolate the effects of domain diversity and class exposure on CLIP's generalization performance.

**Methods And Evaluation Criteria:**

yes

**Other Comments Or Suggestions:**

Nothing

**Other Strengths And Weaknesses:**

Adv:
1.By constructing a variety of controllable training data distributions, the performance of CLIP in domain generalization and combinatorial generalization is systematically studied, and the generalization ability of CLIP is further revealed.
2.The consistency of the experimental results is verified on different model architectures and a large number of data, which enhances the reliability and universality of the conclusions.
3.The key effects of domain diversity and class exposure on CLIP generalization ability are revealed, especially the discovery that partial class exposure may weaken the combinational generalization ability. Four valid CLIP insights are proposed based on experiments.

Dis:
1.The paper mainly focuses on the generalization ability of CLIP model, and does not compare with other similar visual-language models (such as ALIGN, BLIP, etc.). This makes it difficult for readers to fully understand CLIP's strengths and weaknesses in domain generalization and portfolio generalization.
2.Despite the use of multiple datasets, such as DomainNet, these datasets may still be deficient in domain diversity and category richness, so the size and quality of the dataset may also have an impact on the generalization ability of the model, but this is less discussed in the article.
3.The paper mainly uses top-1 accuracy rate as the evaluation index. It is suggested to introduce more evaluation indicators, such as top-5 accuracy, F1 score, etc., to evaluate the performance of the model more comprehensively.
4.The current experiments focus on ImageNet-Captions and DomainNet datasets. It is recommended to extend the experiment to more datasets, especially those containing more diverse fields (such as LAION), to verify the generality of the conclusions.

**Questions For Authors:**

Nothing

**Relation To Broader Scientific Literature:**

The generalization ability of CLIP under different conditions is systematically studied in this paper.

**Theoretical Claims:**

This paper puts forward four findings based on the experimental results, which are reasonable to infer and have sufficient evidence

---

> ### Author Rebuttal · Authors · 2025-03-31
>
> We thank the reviewer for the detailed and constructive feedback. Below we address remaining concerns.
>
> > W1: Comparisons with other vision-language models (VLMs)
>
> We focused on CLIP due to its wide adoption and use in prior related work (e.g., [1, 2]).
>
> We had already verified the consistency of our results with SigLIP (which became more popular recently) in Figure 11 in Appendix B.1. As suggested, we additionally verified consistency of results with BLIP (which are also consistent). We chose to omit ALIGN due to its close similarity to CLIP.
>
> > W2: Discussion of dataset size and quality
>
> We agree that both dataset size and especially quality influence generalization. We will include this to our discussion.
>
> Regarding dataset size: While Figure 10 in Appendix B.1 shows that large base datasets seem to improve generalization performance, there remains a gap to higher diversity settings. We would like to note that the higher generalization performance on CC12M may also be a confounding factor of a less clean and, thus, more diverse base dataset (see the main finding of [2]). As a result, the performance improvement when adding the diverse DomainNet samples is likely smaller because of this.
>
> Regarding quality: Unfortunately, quality remains only loosely addressed in much of the current literature (e.g., [3]), since it is not clear how to quantify it best. Recently, Schrodi et al [4] showed that an information imbalance leads to the modality gap. When we interpret information imbalance as one aspect of data quality, we might be able to indirectly measure this aspect of quality via the modality gap. However, we leave further investigation on how to quantify data quality for future work.
>
> > W3: Additional evaluation metrics
>
> As suggested, we additionally evaluated with balanced top-5 accuracy and multi-class F1 score. The results are consistent with those based on balanced top-1 accuracy.
>
> > W4: Extensions to additional datasets beyond ImageNet-Captions and DomainNet
>
> Figure 10 in Appendix B.1 provides consistent results for the more diverse CC3M & CC12M datasets. However, CC3M & CC12M comprise images from a mix of domains, which creates confounders. Thus, for controlled experiments and clearer analysis, we primarily relied on the cleaner ImageNet-Captions dataset, following prior work [1]. The uncontrolled mix of domains is even more pronounced in the LAION dataset. Verifying our findings on LAION would therefore require robust automated class & domain annotation/filtering methods (which is technically challenging in itself) and large computational resources (which we lack).
>
> ---
>
> [1] Fang, Alex, et al. "Data determines distributional robustness in contrastive language image pre-training (clip)." ICML 2022.
>
> [2] Mayilvahanan, Prasanna, et al. "In search of forgotten domain generalization." ICLR 2025.
>
> [3] Nguyen, Thao, et al. "Quality not quantity: On the interaction between dataset design and robustness of clip." NeurIPS 2022.
>
> [4] Schrodi, Simon, et al. "Two effects, one trigger: on the modality gap, object bias, and information imbalance in contrastive vision-language models." ICLR 2025.

---

> > ### Comment · Reviewer_H9Z3 · 2025-04-05
> >
> > The author answered my questions well and solved my doubts, so I finally gave 4.

---

### Official Review · Reviewer_jP1e · 2025-03-11

**Overall Recommendation:** 3

**Summary:**

This paper tries to find what factors affect the domain generalization and compositional generalization of CLIP. The empirical experiment show that domain diversity is essential for both domain and compositional generalization.

**Claims And Evidence:**

I'm not convinced that the experiments alone back the claims. CLIP's pretraining data is massive and very diverse, so it's unclear if the results are truly due to domain diversity or just the sheer size of the dataset. Additionally, the paper offers no theoretical evidence to support its claim.

**Essential References Not Discussed:**

Include two related works in the introduction—one that enhances domain diversity [1] and another that develops domain-invariant representations in CLIP [2].

[1] Yu, Xi, Shinjae Yoo, and Yuewei Lin. "CLIPCEIL: Domain Generalization through CLIP via Channel rEfinement and Image-text aLignment." Advances in Neural Information Processing Systems 37 (2024): 4267-4294.

[2] Bose, Shirsha, et al. "Stylip: Multi-scale style-conditioned prompt learning for clip-based domain generalization." Proceedings of the IEEE/CVF Winter Conference on Applications of Computer Vision. 2024.

**Experimental Designs Or Analyses:**

One issue is that DomainNet is a long-tail dataset, which may affect the claim.

**Methods And Evaluation Criteria:**

Yes.

**Other Comments Or Suggestions:**

Figure 1A is confusing; please consider redrawing it and replacing the abstract symbols with actual image samples.

**Other Strengths And Weaknesses:**

N/A

**Questions For Authors:**

N/A

**Relation To Broader Scientific Literature:**

Actually, improve domain diversity or learning shared information which could improve domain generalization have already been widely investigated in prior work.

**Theoretical Claims:**

N/A

---

> ### Author Rebuttal · Authors · 2025-03-31
>
> We thank the reviewer for the feedback. Below, we try to address the concerns. However, the review has been very brief, and some points remained unclear to us. We would greatly appreciate further clarification during the discussion phase so we can fully address them, if we have not already done so below.
>
> > Are the results due to domain diversity or just the sheer size of the dataset? (our interpretation of the reviewer’s concern)
>
> To disentangle the effects of dataset size and domain diversity (two typically entangled factors), we conducted controlled experiments, where we fixed dataset size while varying domain diversity. This setup allows us to isolate the impact of diversity. For example, we observe that adding a single domain yields often negligible gains, while greater domain diversity significantly improves generalization performance (see Fig. 2).
>
> In addition, recent work by Mayilvahanan et al [1] showed that “domain contamination [in web-scale datasets] contributes substantially to CLIP’s strong [generalization] performance”  [1, p. 9]. Our work goes beyond theirs by analyzing when and how different domain mixtures influence domain and compositional generalization, e.g., we investigate why CLIP sometimes fails to generalize even with high diversity, see our (mechanistic) analysis in Sec. 6.2.
>
> > No theoretical evidence to support its claim
>
> We would like to clarify that this is an empirical analysis paper, with our claims grounded in controlled and well-motivated experiments – as also noted positively by both other reviewers. While we do not offer theoretical analysis, we follow the growing body of work that investigates model behavior through carefully controlled experiments, such as the popular “Physics of Language Models” series (https://physics.allen-zhu.com/).
>
> > DomainNet is a long-tail dataset, which may affect the claim
>
> DomainNet is long-tail; yet this mirrors the nature of web-scale datasets like LAION-400M (e.g., see Fig. 1a in [2]). As such, we view this as a strength of our experimental setup, which aims to closely replicate CLIP’s training while allowing for controlled dataset manipulations.
>
> To mitigate the effect long-tail in the evaluations, we report *balanced* accuracy throughout (as noted positively by reviewer jK9M). Following reviewer H9Z3’s suggestion, we also verified consistency of our results for additional metrics, such as (standard) accuracy or F1 score.
>
> > Actually, improve domain diversity or learning shared information which could improve domain generalization have already been widely investigated in prior work.
>
> Domain diversity has been broadly studied, but we respectfully note that to the best of our knowledge, ours is the first systematic analysis of domain and compositional generalization in CLIP under OOD conditions. Besides this, we (i) analyzed the impact of partial class exposure on compositional generalization (Finding 2), and/or (ii) conducted mechanistic analyses demonstrating that feature and circuit sharing are needed for generalization (Findings 3 & 4). We would be grateful if the reviewer could provide references that have already explored this.
>
> The references mentioned by the reviewer in “Essential References Not Discussed” enhance CLIP’s domain generalization via adapters [3] or style projectors [4]. However, they primarily focus on method improvements and do not offer in-depth analysis (as also positively noted by reviewer jK9M).
>
> > Include related works
>
> We will add both references [3,4].
>
> > Figure 1A is confusing; please consider redrawing it and replacing the abstract symbols with actual image samples.
>
> We used abstract symbols in Figure 1A to compactly highlight the difference between class sets $C_1$​ and $C_2$. We believe that actual image samples may obscure this important distinction. That said, if other reviewers share this concern, we are happy to revise the figure accordingly.
>
> ---
>
> [1] Mayilvahanan, Prasanna, et al. "In search of forgotten domain generalization." ICLR 2025.
>
> [2] Wen, Xin, et al. "What Makes CLIP More Robust to Long-Tailed Pre-Training Data? A Controlled Study for Transferable Insights." NeurIPS 2024.
>
> [3] Yu, Xi, Shinjae Yoo, and Yuewei Lin. "CLIPCEIL: Domain Generalization through CLIP via Channel rEfinement and Image-text aLignment." NeurIPS 2024.
>
> [4] Bose, Shirsha, et al. "Stylip: Multi-scale style-conditioned prompt learning for clip-based domain generalization." WACV 2024.

---

> > ### Comment · Reviewer_jP1e · 2025-04-04
> >
> > Thank you so much for your detailed response, it has addressed most of my concerns. However, I’m still a bit confused by Figure 1. Could you please clarify what the different colors represent and what the various shapes of the abstract symbols indicate? Specifically, in Figure 1A, which section corresponds to the training set and which to the test set? Additionally, references [3,4] appear to enhance CLIP’s generalizability by incorporating a more diverse range of style representations and emphasizing invariant features, which aligns well with your findings. I believe it’s worth mentioning that prior work has already improved CLIP’s generalizability in this way.

---

> > > ### Author Response · Authors · 2025-04-04
> > >
> > > Thank you very much for your positive feedback. We are happy to hear that our previous response addressed most of your concerns. Below, we aim to clarify your remaining questions regarding Figure 1 and how we plan to incorporate references [3,4].
> > >
> > > > Clarification on Figure 1
> > >
> > > In Figure 1, the *colors* represent different domains (red = natural domain; blue = test domain; green/orange = additional training domains) and the *shapes* represent the two distinct class subsets (square = $c_1,...,c_k=C_1$; circle = $c_{k+1},...,c_n=C_2$). Note that we need these subsets for testing compositional generalization, where some classes of the test domain are seen during training (the squares) and others are not (the circles).
> > >
> > > The *opacity* of each symbol indicates whether a class subset from a domain (e.g., blue squares) is included (opaque) or excluded (more transparent) from the training set. Inclusion or exclusion depends on the specific setup, e.g., in the natural-only setup, only natural images from both class subsets are included in training.
> > >
> > > Across *all* setups, the *test set* corresponds to the blue circles (see also Figure 1B), which are held out from training. All other colored shapes, including blue squares, may be included in the training, depending on the specific setup.
> > >
> > > We will incorporate these clarifications into Figure 1’s caption to improve its clarity.
> > >
> > > > Incorporation of references [3,4]
> > >
> > > We agree that both works make significant contributions toward enhancing CLIP’s generalization through diverse styles or invariant features. We will therefore add these references, as well as those suggested by reviewer jK9M in “Relation to Broader Scientific Literature”. We will clarify how our analysis extends these works by offering a *more detailed analysis* of the factors that enable generalization - and, crucially, why generalization may *still fail*.
> > >
> > > > Final note
> > >
> > > We hope that our discussion has addressed your concerns and questions. We would greatly appreciate it if you could consider updating your recommendation to reflect your assessment of the paper following our constructive exchange. Thank you again for your valuable feedback - it has helped us strengthen our work.

---

### Official Review · Reviewer_jK9M · 2025-03-17

**Overall Recommendation:** 4

**Summary:**

The paper studies the generalization capabilities of CLIP in the Domain Generalization setting. Specifically, the authors study when and how clip exhibits domain generalization - when a model generalizes to unseen domains, and compositional generalization - when a model generalizes to classes from partially seen domains during training. To facilitate this study, the authors conduct carefully crafted experiments on CLIP with DomainNet and ImageNet to understand the influence of domain diversity, language descriptions, and shared representations on the generalization capabilities of CLIP.

**Claims And Evidence:**

Yes. All the claims made in the submission are supported by well-motivated and clear experiments.

**Essential References Not Discussed:**

The authors have referenced the relevant literature and discussed the same adequately.

**Experimental Designs Or Analyses:**

- **General setups:** The authors consider two experimental setups: domain generalization - where the model is required to generalize to an unseen domain, and compositional generalization - where the model has partial access to classes from the unseen target domain. The authors train CLIP in both of these settings by merging data from ImageNet-Captions and DomainNet. These experimental designs follow the general setup of generalization works.
- **Experiments on the role of visual embeddings:** In Sec. 6.1, the authors discuss their experimental setup and results in determining the role of the shared visual embeddings across domains in facilitating generalization in CLIP. Specifically, the authors train a Sparse Autoencoder (SAE) on the CLIP visual representations across domains and consider the overlap between the top-k embeddings across domains. However, the details of this experiment are unclear (please see the weaknesses section below).

Overall, the experiments and analyzes presented in the submission are valid for the study of CLIP's generalizability across domains.

**Methods And Evaluation Criteria:**

The proposed method and evaluations use relevant benchmark datasets, and the authors have justified their choice of datasets and evaluation settings.

- **Datasets**: For their experimental evaluation, the authors have chosen ImageNet-Captions, ConceptualCaptions 3M, or ConceptualCaptions 12M as their base dataset, along with domains from the DomainNet dataset for the domain-specific image-text pairs. Given that DomainNet remains the largest domain-shift benchmark for domain generalization, this choice is verified.
- **Evaluation metrics**: The authors use balanced top-1 accuracy as their evaluation metric, which mitigates the effect of the long-tail nature of the DomainNet dataset (L192-195).

Similarly, authors have clearly outlined and justified their setups for evaluating the effect of shared representations on DG performance, and the behavior of CLIP on domains such as quickdraw via model circuit similarities across pairs of domains.

**Other Comments Or Suggestions:**

Some of the tables and their captions are unclear and can be better presented for enhanced readability. For example, Tables 3 and 4 do not mention the quantity being presented (whether it is the improvement in accuracy or the accuracy itself). Additionally, as mentioned in the weaknesses section, some experimental settings have not been clearly explained. The reviewer suggests the authors review these details and rewrite them for better understanding.

**Other Strengths And Weaknesses:**

### Strengths
- **Relevance to prior works:** The major strength of this submission is its relevance to prior works that utilize pre-trained Vision-Language Models (VLMs) such as CLIP for domain generalization. Specifically, as mentioned above, most works in DG directly consider the generalizability of CLIP as a given and design frameworks using CLIP while the submission delves deeper into the mechanics behind CLIP's generalization capabilities. A noteworthy contribution is the submission's study on the quickdraw domain from DomainNet, which prior works often dismiss as unseen by CLIP.
- **Experiments and results:** The submission presents well-motivated and simple experiments to analyze the properties of CLIP in the domain and compositional generalization settings. Specifically, the experiments on the role of shared intermediate features and similarity between model circuitry across domains is quite interesting.
- Overall, the paper is well-written, easy to follow and understand. All the experiments are well-motivated, outlined clearly and analyzed well.

### Weaknesses
(a) As mentioned above, while the authors have presented an experiment to substantiate the claim that generalizable models share representations across domains, the experiment itself is unclear.
- How is CLIP evaluated using this trained SAE? Are these evaluations conducted in the leave-one-out or CG settings? Which pairs of domains are considered in the setup, i.e., all domains or only the training domains?
- How are the top-k features chosen across domains? What is the objective used to compute the top-k features (eg: cosine similarity, L2 loss, etc.)?
- How is the performance delta collected (in other words, how do the authors evaluate the top-k representations to present the improvement using the shared embeddings)?

**Questions For Authors:**

- In Sec. 6.1, the authors present an experiment to analyze the role of shared representations in generalization. Specifically, they train a SAE in the representation space and consider the top-k features. Can the CLIP visual features directly be considered for the top-k computation and further analyzes?

**Relation To Broader Scientific Literature:**

- The contributions of the paper are highly relevant to domain generalization literature. Specifically, several papers utilize the generalization capabilities of CLIP [1, 2, 3] for a typical domain generalization without delving into the analysis of CLIP's generalizability.
- Moreover, the general consensus for the lack of performance of CLIP on the Quickdraw domain in DomainNet has largely been attributed to the lack of images from this domain in the pre-training data of CLIP. However, there has been no work that studies the behavior of CLIP with such domains, which this work explores in great detail.

1. Addepalli, Sravanti, et al. "Leveraging vision-language models for improving domain generalization in image classification." CVPR 2024.
2. Huang, Zeyi, et al. "A sentence speaks a thousand images: Domain generalization through distilling clip with language guidance." ICCV 2023.
3. Yu, Xi, Shinjae Yoo, and Yuewei Lin. "CLIPCEIL: Domain Generalization through CLIP via Channel refinement and Image-text alignment." NeurIPS 2024.

**Theoretical Claims:**

The paper is an empirical study of the generalization properties of CLIP. As a result, there are no significant theoretical claims or proofs in the submission.

---

> ### Author Rebuttal · Authors · 2025-03-31
>
> We thank the reviewer for the insightful and constructive feedback. Below, we address the remaining concerns and questions.
>
> > Can the CLIP visual features directly be considered for the top-k computation and further analyzes?
>
> Yes, this is possible. However, we would like to note that CLIP’s visual features are in superposition and not directly interpretable w.r.t. an object class. For example, Schrodi et al [1] showed that some of CLIP’s visual feature dimensions have to do something with the semantically non-meaningful modality gap. Thus, recent work [2] and ours opt for dictionary learning methods like SAEs (i.e., linear vectors that need not be axis-aligned) to extract more interpretable and class-relevant features.
>
> > Missing details about the SAE experiment (including the questions raised by the reviewer)
>
> We thank the reviewer for pointing this out. Below, we provide a detailed explanation of our SAE experiment, which we will add to the paper.
>
> We trained a separate SAE on CLIP’s visual features for each of the following 5 models: a natural-only baseline, two leave-out-domain models (where either clipart or sketch is the test domain), and two CG high-diversity setting (also using clipart or sketch as test domain).
>
> For each model, we analyzed the extent to which the top-k most activating SAE features are shared between the test domain (clipart or sketch) and all other domains (not just the training domains). To identify the top-k SAE features, we computed the SAE hidden representations (after applying the non-linearity) for each sample of each class and domain. We then selected the top-k SAE features (with $k \in \lbrace 5, 10, 15, 20\rbrace$) per class-domain pair based on how frequently they ranked among the top-20 most activating features (i.e., largest activation magnitudes).
>
> To measure feature sharing, we calculated the percentage overlap between the top-k SAE features of the test domain (clipart or sketch) and those of each of the other domains using the top-k SAE features that we got from above. To yield a single overlap score per model, we averaged across classes, these pairs of domains, and the four values of k. Finally, we calculated the deltas in Table 3 by comparing the overlap score of the natural-only baseline model with the corresponding leave-out-domain and CG high-diversity models.
>
> > False positives of Weisfeiler-Lehman (WL) kernel
>
> Thank you for pointing this out. We will add this aspect.
>
> While the WL kernel can yield false positives, this is unlikely in our case: The node names in the circuits encode global topological information (i.e., layer and neuron indices), which reduces the chance of false positives. Besides that, we employed the 3-WL kernel, which is more powerful than, e.g., 1-WL, and also reduces the chance of false positives.
>
> Besides that, we verified that the graphs were distinguishable at the node level (0-th iteration of the WL kernel); as indicated by Figure 6b (the figure just shows aggregated results, but we verified it for each set of nodes of each pair of circuits). Note that false positives can only occur if two graphs remain indistinguishable across all iterations of the WL kernel, which is empirically not the case.
>
> > Revise some of the tables, captions, and experimental settings
>
> We will revise them.
>
> ---
>
> [1] Schrodi, Simon, et al. "Two effects, one trigger: on the modality gap, object bias, and information imbalance in contrastive vision-language models." ICLR 2025.
>
> [2] Rao, Sukrut, et al. "Discover-then-name: Task-agnostic concept bottlenecks via automated concept discovery." ECCV 2024.

---

> > ### Comment · Reviewer_jK9M · 2025-04-05
> >
> > Thank you for your response. The points discussed in the rebuttal have addressed all of my concerns. Thus, I maintain my score at 4.

---

### Decision · Program_Chairs · 2025-05-01

**Decision:**

Accept (spotlight poster)

**Comment:**

This paper studies the generalization behavior of CLIP by varying training distribution and controlling diversity and object class exposure. The paper looks at domain generalization and compositional generalization, and shows that domain diversity is essential for both. This topic is of wide interest (generalizability of CLIP), and the paper includes a large set of experiments to provide more understanding about the limitations of generalization of CLIP. The paper also presents a mechanistic analysis suggesting that compositional generalization requires sharing of intermediate features and circuits. The reviewers have raised a few issues that the authors have addressed convincingly. The authors are requested to make relevant modification in the final version of the paper.